

# In situ measurements of meltwater flow through snow and firn in the accumulation zone of the SW Greenland Ice Sheet

Nicole Clerx[1], Horst Machguth[1], Andrew Tedstone[1], Nicolas Jullien[1], Nander Wever[2], Rolf Weingartner[3], and Ole Roessler[3,a]

[1]Department of Geosciences, University of Fribourg, Fribourg, Switzerland
[2]Department of Atmospheric and Oceanic Sciences, University of Colorado Boulder, Boulder, CO, USA
[3]Institute of Geography and Oescher Centre for Climate Change Research, University of Bern, Bern, Switzerland
[a]now at: German Federal Institute of Hydrology (BfG), Koblenz, Germany

**Correspondence:** Nicole Clerx (nicole.clerx@unifr.ch)

**Abstract.** The Greenland Ice Sheet is losing mass, part of which is caused by increasing runoff. The location of the runoff limit, the highest elevation from which meltwater finds its way off the ice sheet, plays an important role in the surface mass balance of the ice sheet. The recently observed rise in runoff area might be related to an increasing amount of refreezing: ice layer development in the firn hinders vertical percolation and promotes lateral runoff. To investigate meltwater flow near

the runoff limit in the accumulation zone on the southwest Greenland Ice Sheet, we carried out *in situ* measurements of hydrological processes and properties of firn and snow. The hydraulic conductivity of icy firn in pre-melt conditions measured using a portable lysimeter ranges from 0.17 to 12.8 m hr[-1], with flow predominantly occurring through preferential flow fingers. Lateral flow velocities of meltwater on top of the near-surface ice slab at the peak of the melt season measured by salt dilution- and tracer experiments range from 1.3 to 15.1 m hr[-1]. With these lateral flow velocities the distance between the slush limit, the

highest elevation where liquid water is visible on the ice sheet surface, and the runoff limit could be up to 4 km in regions where near-surface ice slabs are present. These measurements are a first step towards an integrated set of hydrological properties of firn on the SW Greenland Ice Sheet, and show evidence that meltwater runoff might occur from elevations above the visible runoff area.

## 1 Introduction

Since 1991 the Greenland Ice Sheet (GrIS) has lost around 4000 gigatonnes of mass, which corresponds to roughly 10 mm of sea level rise (the IMBIE Team, 2019). Over a third of this mass loss, ∼34%, is accounted for by a negative surface mass balance (Mouginot et al., 2019). Meltwater runoff, one of the major surface mass balance parameters, has increased by >40% since the 1990's due to a warming climate (Hanna et al., 2012; Hall et al., 2013). This has caused the contribution from the GrIS to global mean sea level rise to increase from <5% in 1993 to >25% in 2014 (Chen et al., 2017).

Quantifying where and why runoff takes place, i.e. what governs the location and evolution of the runoff limit throughout the melt season, is critical for accurate firn modelling and ice sheet mass balance (van As et al., 2016; Nienow et al., 2017). Even though only meltwater that runs off contributes to mass loss of the GrIS, estimates of refreezing and retention of melt



as predicted by climate- and SMB models currently are subject to high uncertainties (Smith et al., 2017; Nienow et al., 2017). Existing parametrisations that are used for firn densification and vertical meltwater percolation in Greenland-wide firn models

(e.g. Brown et al., 2012; Marchenko et al., 2017; Steger et al., 2017), however, are not uncommonly based on knowledge gained from other environments such as seasonal snowpacks and/or smaller (alpine or arctic) glacier settings, and actual conditions of meltwater runoff to occur on the GrIS remain largely unvalidated.

Firn has a large buffering capacity for meltwater through refreezing in its pore space (Pfeffer et al., 1991; Harper et al., 2012), and covers over 80% of the GrIS (Box et al., 2012; Fausto et al., 2018). According to Greenland-wide modelling of

meltwater retention using current knowledge and parametrisations, approximately 45% of the generated meltwater has been retained in firn over the past five decades (van Angelen et al., 2013; Noël et al., 2016; Steger et al., 2017). Firn structure is therefore an influential parameter in the surface mass balance of the GrIS (van den Broeke et al., 2017).

In recent years, in particular after extreme melt events, firn stratigraphy in the accumulation zone of the GrIS has changed significantly. *In situ* observations as well as ground-based and airborne radar data show that widespread near-surface ice slabs

have rapidly developed and expanded to higher elevations (Machguth et al., 2016; MacFerrin et al., 2019; Culberg et al., 2021). These ice slabs reduce the overall buffering capacity of the firn layer by impeding meltwater from percolating into the porous, underlying firn (Nghiem, 2005; Humphrey et al., 2012; Polashenski et al., 2014) and hence could force large amounts of water to run off at the surface (de la Peña et al., 2015). Especially on the southwestern GrIS, where widespread superficial ice slabs are present, meltwater can potentially travel long distances before ponding in supraglacial lakes and/or draining through moulins

to the ice sheet bed (Chu, 2014).

In the south-west of the GrIS, recent ice slab formation has enabled meltwater to run off in supraglacial rivers from elevations at least as high as 1840 m a.s.l. (Machguth et al., 2016). These rivers tend to initiate in slush fields: water-saturated areas of firn and snow with visible surface ponding (Chu, 2014). We define the slush limit as the uppermost altitude at which liquid meltwater is visible at the surface during the melt season, as first suggested by Müller (1962). This term was later added to

Benson's widely used firn classification scheme (1996). The runoff limit is the elevation below which meltwater leaves the ice sheet. We hypothesize that the runoff limit generally lies above the slush limit. The hydrological processes occurring at the slush- and runoff limit are critical for meltwater retention and runoff, but to what extent these two limits are related and how they are affected by changes in firn stratigraphy (i.e. meltwater refreezing) is unclear.

To date, the location of the slush- and runoff limit has received relatively limited attention. The slush limit has been mapped

using AVHRR satellite data by Greuell and Knap (2000) and using SMB analyses (Reeh, 1991), but these studies have triggered limited further investigations.As we lack adequate constraints on the hydrological properties of snow and firn in the GrIS accumulation area, it is currently hard to quantify meltwater retention and -runoff, and to predict where this occurs.

We focus on the hydrological processes in, and matrix properties of snow, slush and firn near the runoff limit on the southwest Greenland Ice Sheet. We undertook two distinct fieldwork campaigns to measure the hydrology of snow, slush and firn. In one

campaign we investigated the hydrological properties of icy firn above the current runoff limit, with an emphasis on constraining vertical percolation rates. In the other campaign we examined a summer-time slush field overlying an impermeable ice slab, focusing on its ability to transport meltwater laterally. Here we present measured values for vertical percolation velocities





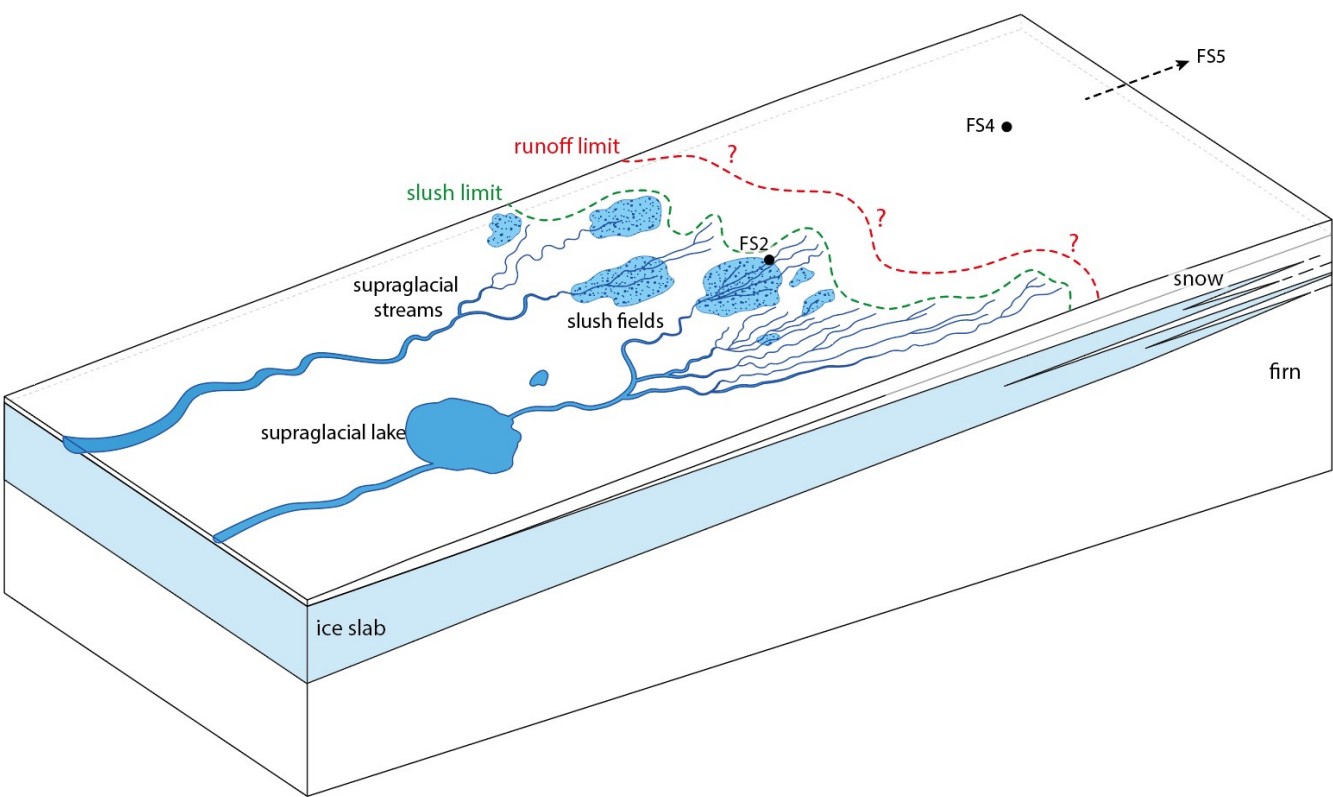

**Figure 1.** Schematic overview of the elements of the hydrologic system in the accumulation zone of the SW GrIS. Water percolating through the snow/firn pools into slush regions and eventually is evacuated through supraglacial streams. The exact location of the runoff limit is unknown, since liquid meltwater can be present above the slush limit during the summer season. The FS2, FS4 and FS5 labels indicate the approximate position of measurement sites in this study.

through icy firn as well as measurements of lateral meltwater flow velocity through water-saturated slush directly on top of near-surface ice slabs. We furthermore present measured values for snow and firn permeability and compare these to other
existing literature data. Lastly, we show how our measured values compare to these parameters when calculating them based on existing parametrisations.

## 2 Field site description

The study area (Fig. 2) is located in the southwestern part of the Greenland Ice Sheet, around the upper end of the K-transect (van de Wal et al., 2005) which is a region with an excellent availability of firn records (Rennermalm et al., 2021). The field
sites (Fig. 1, Table A1) are in the accumulation area near the elevation at which, in recent years, the slush limit occurred.





The slush limit coincides with the location where recent widespread near-surface ice slabs have been identified in this area (Machguth et al., 2016; MacFerrin et al., 2019). Fieldwork was carried out in July-August 2020 and in April-May 2021.

During summer fieldwork in 2020, the slush limit was not as high as shown in Fig. 2a: at the start of the field campaign the slush limit was clearly below the elevation of the FS2-site, based on observations in the field and confirmed by Sentinel data. During the field campaign, liquid water presence in the area occurred at progressively higher elevations (Fig. 2b).

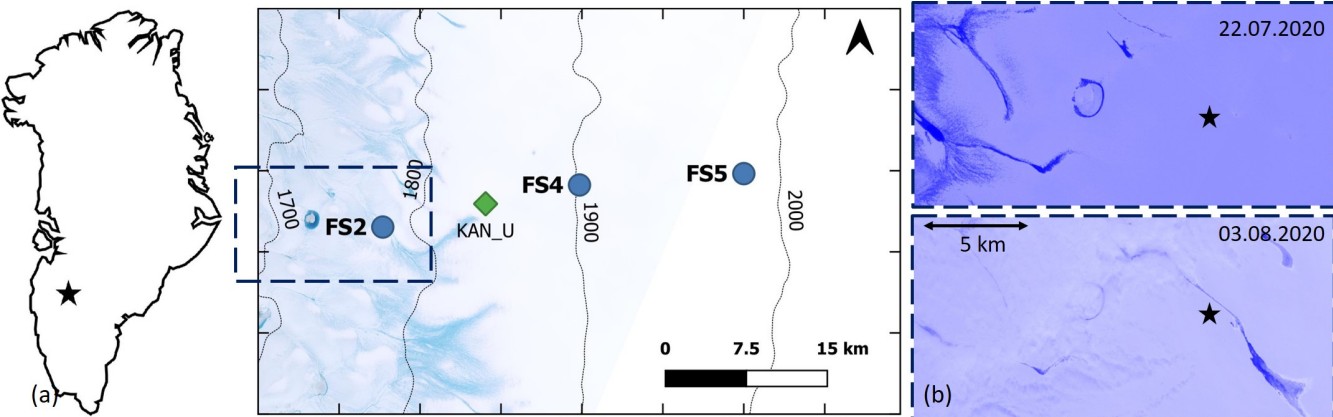

**Figure 2.** (a) Location map of the study area, showing the study sites on the Greenland Ice Sheet (FS2 for summer measurements, FS4 for spring data collection, both sites and FS5 for firn stratigraphy and KAN_U for meteorological data). Thin black lines represent elevation contours (m a.s.l.) from the ArcticDEM modified to show elevation in m a.s.l. (Porter et al., 2018). The background image is a Sentinel 2 true color composite from 12.08.2019, around the time of maximum melt extent that year. The dashed dark blue rectangle indicates the outline of the composites shown in panel b. (b) Sentinel-2 NDWI composites showing the liquid water presence on the ice sheet surface around the start and end of the summer 2020 field campaign (source: sentinelhub Playground). The black star indicates the location of field site FS2. Note that the NDWI composites have not been corrected for cloud artefacts.

## 3   Theoretical background

Firn and snow are mixtures of air, ice and water, in which metamorphic processes that change the morphology and physical properties of the snow particles play an important role. The density of snow or firn $\rho$ is related to the density of air $\rho_a$, the density of water $\rho_w$ and the density of ice $\rho_i$:

$$\rho = \rho_a \theta_a + \rho_w \theta_w + \rho_i \theta_i \tag{1}$$

where $\theta_a$, $\theta_w$ and $\theta_i$ are the air porosity, liquid water content and ice fraction, respectively.

Porosity $\phi$ is the volume fraction of pore space in a medium. When neglecting the density of air and assuming that the volume fraction of water $V_w \approx 0$ it can be calculated as:

$$\phi = \frac{V_a + V_w}{V_T} \simeq 1 - \frac{\rho}{\rho_i} \tag{2}$$





where $V_a$ is the volume fraction of air, $V_T$ is the total volume [m³], $\rho$ is the sample density [kg m⁻³] and $\rho_i$ is the density of ice [kg m⁻³].

The liquid water content $\theta_w$ and liquid water saturation $S_w$ are two distinct measurements of snow wetness: $\theta_w$ is defined as a volumetric percentage of total volume, whereas water saturation $S_w$ is defined as the amount of pore space occupied by liquid water:

$$S_w = \frac{\theta_w}{\phi} \tag{3}$$

We define slush as snow/firn in which all pore space is occupied by liquid water, i.e. $S_w = 1$.

The irreducible water saturation $S_{w,ir}$ is the residual fraction of liquid water that cannot be removed from the pore space due to capillary forces. According to several experimental studies (e.g. Colbeck, 1974; Coléou and Lesaffre, 1998) the irreducible water saturation $S_{w,ir}$ of snow is approximately 7%. Measured values of LWC in a ripe snowpack have been reported between 2% and 4% (e.g. Jordan et al., 1999; Yamaguchi et al., 2012; Katsushima et al., 2013).

Since meltwater flow through snow and firn is analogous to flow through a porous medium, the empirical Darcy's law can be used to calculate the hydraulic conductivity of firn:

$$q = \frac{Q}{A} = -K\frac{\partial h}{\partial z} = -\frac{k}{\mu}\frac{\partial h}{\partial z} \tag{4}$$

where $q$ is the instantaneous flux [m s⁻¹], $Q$ is discharge [m³ s⁻¹], $A$ is the area through which flow occurs [m²], $K$ is hydraulic conductivity [m s⁻¹], $k$ is the permeability of the medium [m²], $\mu$ is the dynamic viscosity [Pa·s] of the fluid and $\partial h/\partial z$ is the hydraulic gradient. The hydraulic gradient describes the difference in hydraulic head $h$ [m], which is defined as:

$$h = \Psi + z = \frac{P}{\rho g} + z \tag{5}$$

where $\Psi$ is the pressure head [m], $z$ is the elevation head [m], $P$ is the fluid pressure [Pa], $\rho$ is bulk density of the fluid [kg m⁻³] and $g$ is acceleration due to gravity [m s⁻²].

Permeability is an intrinsic material property that indicates the ability for fluids to flow through this material, independent of the fluid. It is a function of porosity, but also related to grain shape and connectivity of the pores. The hydraulic conductivity of a porous medium for fluid flow is related to permeability as follows:

$$K = k\frac{\rho g}{\mu} \tag{6}$$

Re-writing equation 4 for vertical percolation to solve for hydraulic conductivity gives, since $\partial h/\partial z = 1$ in this case:

$$-K = \frac{Q}{A} \tag{7}$$

For the case of lateral meltwater flow through a porous medium without a significant pressure drop, the hydraulic gradient reduces to the elevation head $\partial h/\partial L$ only and hence Darcy's law can be re-written as:

$$-K = \frac{k\rho g}{\mu}\frac{\Delta H}{\Delta L} \tag{8}$$



where $\Delta H$ is the elevation difference [m] over a distance of $\Delta L$ [m].

The Kozeny-Carman equation relates a medium's permeability to pressure drop and fluid viscosity for laminar flow through a packed bed of solids, and when combined with Darcy's law it can be used to predict permeability (Kozeny, 1927; Carman, 1937; Bear, 1972):

$$k = \epsilon_s^2 \frac{\phi^3 D_p^2}{150(1-\phi)^2} \tag{9}$$

where $\epsilon$ is sphericity (= 1 for perfect spheres and between 0 and 1 for all other grain shapes), $\phi$ is porosity and $D_p$ is effective
grain diameter [m].

For snow and firn, many other parametrisations exist that relate permeability to density or grain size, either based on direct measurements of air- or liquid permeability (e.g. Shimizu, 1970; Jordan et al., 1999; Albert et al., 2000), or on numerically computed material properties using 3-D microstructural images (e.g. Freitag et al., 2002; Calonne et al., 2012). Since the power law exponents in parametrisations based on direct permeability measurements are commonly site-specific (Adolph and Albert,
2014), here we use the parametrisation by Calonne et al. (2012) that links permeability to specific surface area, density and microstructural anisotropy:

$$k = (3.0 \pm 0.3) r_{es}^2 \exp((-0.0130 \pm 0.0003)\rho_s) \tag{10}$$

where $r_{es}$ is the equivalent sphere radius [m] and $\rho_s$ is snow density [kg m$^{-3}$]. The equivalent sphere radius relates the specific surface area of a snow particle (SSA) to ice density ($\rho_i$) as follows: $r_{es}$ = 3/(SSA·$\rho_i$) (German, 1996).

## 4   Hydrology of icy firn above the runoff limit

### 4.1   Methods

To measure firn density and and stratigraphy, we drilled multiple 13 cm diameter firn cores during the spring 2021 field campaign using a Kovacs coring system. Firn stratigraphy was logged at cm-scale, and 10 cm core sections were measured and weighed for density. Furthermore, traditional snowpack profiles in snowpits were analysed following the recommendation
of Fierz et al. (2009), including observations on grain size and shape, snow temperature, layer thickness, snow hardness and density.

To measure the hydraulic properties of icy firn, i.e. firn interspersed with discontinuous ice lenses, we carried out meltwater percolation experiments. A portable lysimeter, "Rain On Snow Appliance" (ROSA) was installed in a temporary laboratory at research site FS4 (Fig. 2) to investigate water percolation and retention in firn. ROSA was originally designed and constructed
by the University of Bern, and used to study hydrological processes during rain-on-snow events in the Swiss Alps (Probst, 2016; Zaugg, 2017). At the University of Fribourg, the device was optimised for systematic measurements of parameters which are required to determine the hydraulic conductivity and water retention capacity of icy firn.





### 4.1.1 Measurement set-up

ROSA consists of a frame with a square base and a height of about two metres (Fig. 3). The sprinkling system through which
liquid water is delivered to the snow or firn block is attached at the top of the frame. Irrigation intensity is controlled by a
digital Alicat LC flow controller, which has an operating range of flow rates between 0.5 and 500 cm$^3$ min$^{-1}$ and an accuracy
of $\pm 2\%$ of the set flow rate. Liquid water used for simulating melt is dyed (low concentration solution of Rhodamine WT),
and delivered to the sprinkling system from a barrel through Comet submersible aquarium pumps. The firn sample is placed in
a cage which is suspended from the frame with strain-based load cells. The runoff from the firn block is collected below the
cage, and channelled into two Rainwise tipping bucket rain gauges.

To ensure uniform irrigation of the firn block surface, the sprinkling system is moved by an electrical motor. In the field, the
movement motor stopped working after a few experiments. A stationary sprinkling system resulted in deep holes within the
firn block, as liquid water did not disperse homogeneously within the sample. For subsequent experiments the sprinkling head
was moved manually in 2 cm increments every 3 minutes.

At the base, middle and top of the sample cage, 1 Hygroclip and 2 HygroVUE5 sensors are attached to the metal frame to
measure air temperature and humidity during the experiments. The flow controller, tipping buckets, temperature- and TDR-
sensors (Campbell Scientific 107 temperature- and CS655 TDR-probes), and the 3 hygroclips are connected to a Campbell
Scientific CR1000 datalogger that records data at a 10 second interval.

Ambient temperature was monitored during each experiment to prevent melting of the firn sample or refreezing of discharged
water. Whenever the ambient temperature rose above 0°C we used an electrical fan to blow colder outside air into the tent.
Too low temperatures were avoided by doing experiments only at times when solar radiation could sufficiently heat the tent.
Furthermore, the base of the metal plate funneling discharged water into the tipping buckets is equipped with resistor heating
wires to prevent freeze-on.

Before the start of each percolation experiment, sensors were inserted into the firn sample. Four temperature sensors were
inserted to about 20 cm horizontal depth very near (∼1 cm above) the sample base. A fifth sensor was used for permanent
monitoring of the water temperature in the rain barrel from which water was delivered to the sprinkling head. Two TDR-
sensors were installed at the front and back side of the firn block, approximately 20 cm inside the sample and about 10 cm
above its base.

### 4.1.2 Experimental procedure

Snow and firn samples were collected at field site FS4, either from snowpits or from a 2 m-deep 'firn quarry' at the measurement
site. Stratigraphy of the snowpits and the quarry was described following the international classification of seasonal snow on
the ground (Fierz et al., 2009).

Samples were transported and inserted into ROSA manually. Once inside the measurement cage, sensors were drilled into
the sample using a 1 cm diameter, 20 cm long drill bit mounted onto a battery-powered drill. The main experiment started once
the measurement set-up had been initialised according to a pre-experiment checklist for metadata collection. Length, width



**Figure 3.** The Rain On Snow Appliance (ROSA) as deployed on the Greenland Ice Sheet. Items related to the water circulation are labelled in blue, other parts are highlighted in green.

and height of the block were measured close to the sample edges once the experiment had started, taking the average of 3 measurements on each side for height determination.





All experiments were carried out at a fixed flow rate of 100 cm$^3$ min$^{-1}$, which is equivalent to a meltwater supply rate of roughly 12 mm hr$^{-1}$ for this measurement set-up. The exact duration of individual experiments was not fixed beforehand, but we made sure that single experiments lasted long enough for through-flow to occur for around 30–60 minutes (Table 1). Theoretically, longer experiment durations would have been desirable, but this proved unfeasible on the ice sheet, where the meteorological conditions introduced practical and physical limitations. Upon completion of an experiment we recorded preferential flow paths and block stratigraphy with photographs. Starting at the front vertical face of the block we made individual slices with a spacing of 10 cm.

## 4.2 Results

### 4.2.1 Firn stratigraphy

To determine firn stratigraphy and identify the extent of near-surface ice slabs, we drilled a total of 5 firn cores at FS2 (12 m depth), FS4 (5 m and 22 m) and FS5 (5 m and 21 m) – Fig. 4. The thick ice slab visible at FS2 does not extend to FS4. Although ice lenses of multiple meters thick are still present in the cores at FS4, the ice content in the top 10 m of the firn (excluding the seasonal snowpack) has reduced from 94% at FS2 to 54% at FS4. At FS5, which is at an even higher altitude, total ice content in the uppermost 10 m of the firn further decreases to 35% and maximum ice lens thickness is less than 1 m.

### 4.2.2 Vertical water percolation through icy firn

Using ROSA, we carried out 9 meltwater percolation experiments, of which 3 were snow samples and 6 were firn blocks. Samples for *snow1* and *snow2* were collected at the ice sheet surface and consisted of wind-blown snow. The block for *snow3* originated from a snowpit at ~1.5 m depth and was made up of older, transformed, relatively coarse-grained snow including layers of depth hoar, alternated with layers of finer-grained wind-blown snow. All firn samples were dug out from the 2 m-deep quarry close to the laboratory tent. The results of the 7 experiments highlighted in Table 1 are discussed in this section. The initial two experiments (using surface snow) are not further detailed because they served only to familiarize with running ROSA on the GrIS.

All firn blocks contained several discontinuous ice lenses with a thickness ranging between 0.5-2 cm. One firn sample (*firn4*) contained a thicker ice lens of 3-5 cm, visible on all sides of the block. Firn blocks were approximately 0.50 m$^2$ in surface area (roughly 70 by 70 cm wide/long), with a thickness ranging from 13 to 28 cm and an initial density between 414 and 600 kg m$^{-3}$.

Hydraulic conductivity was calculated using Eq. 7 for the final 15 minutes of outflow during each experiment before the water inflow was stopped. Figure 5 shows the evolution of hydraulic conductivity, density and firn sample mass over time for all percolation experiments, and Table 2 shows the main calculated parameters. In all experiments there is a clear shift in the apparent rate of densification once continuous water flow commences. This is especially clear in for example *firn2, firn5* and *firn6* (Fig. 5). The formation of preferential flow paths is probably the reason for these distinct stages of apparent densification rate. Before continuous outflow occurs, the amount of preferential flow paths is insufficient to evacuate the supplied meltwater.



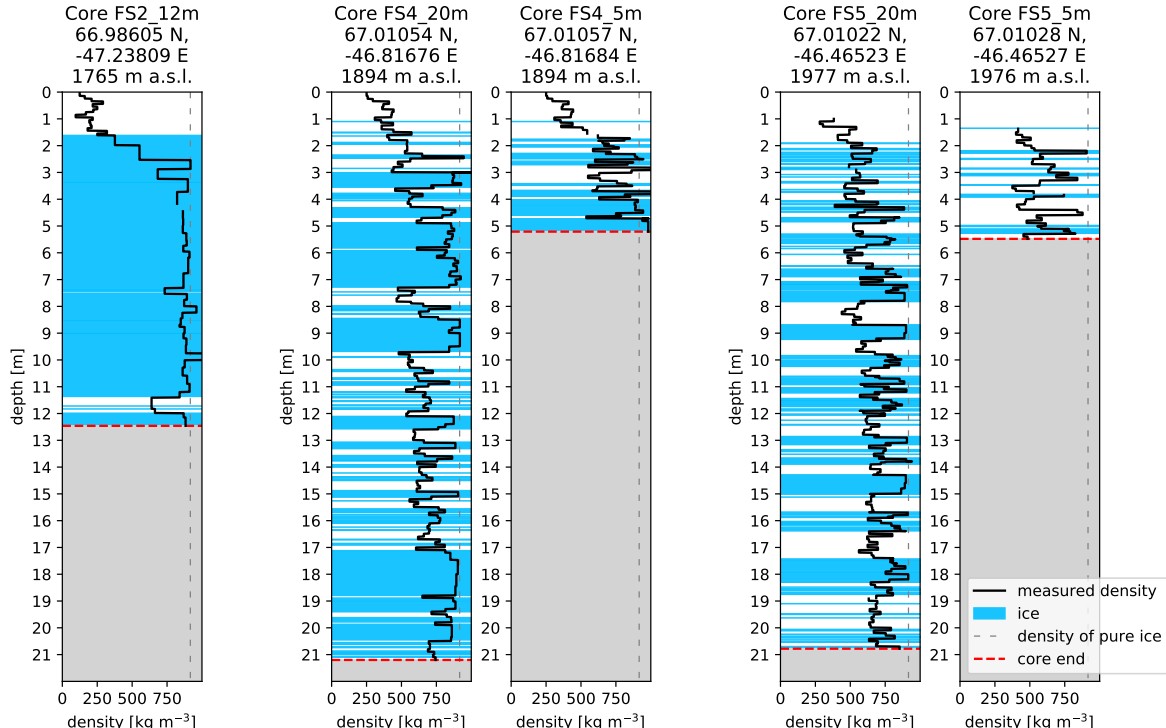

**Figure 4.** Firn stratigraphy at the three field sites as measured in spring 2021. Core logs are displayed W-E or low-high altitude from left to right. Core FS2_12 is at the location of the 2020 summer field measurements, whereas the meltwater percolation experiments carried out in spring 2021 are colocated with Core FS4_20m and Core FS4_5m (see Fig. 2).

Once outflow starts the development of preferential flow paths still continues, until sufficient water evacuation channels have

developed at which the densification rate becomes more or less constant.

Measured hydraulic conductivity values range between 1.71 and 12.80 m hr$^{-1}$ (= 47–356·10$^{-5}$ m s$^{-1}$), with an average of 8.60 ± 3.58 m hr$^{-1}$. Permeability was calculated in two ways: (i) derived from the hydraulic conductivity (Eq. 7), and (ii) calculated using Eq. 6 in combination with Calonne's parametrisation (2012, Eq. 10) . Minimum and maximum permeability of the analysed samples varied between 0.87·10$^{-10}$ and 6.50·10$^{-10}$ m$^2$ according to the Darcy-based calculation. A larger per-

meability range was found using Calonne's parametrisation with an estimated average grain size for each of the firn blocks: 1.61·10$^{-9}$–1.08·10$^{-8}$ m$^2$.

Outflow occurred in all experiments, and experiment duration was based on having a period of at least 30-40 minutes of continuous water percolation through the sample unless we encountered technical problems (see below). Lag times between experiment- and outflow start ranged from 23 minutes to 1 hour and 7 minutes, although no significant relationship

between outflow lag and initial density or any other measured variable exists. Velocities of unsaturated flow were calculated using the lag time and sample height, and range from 0.167–0.438 m hr$^{-1}$ (= 4.65·10$^{-5}$–1.22·10$^{-4}$ m s$^{-1}$), with an average of





**Table 1.** Metadata for the various meltwater percolation experiments. Experiment names in **bold** represent experiments discussed in detail in Sect. 4.2.2

| experiment | sample thickness [m] | initial density [kg m$^{-3}$] | initial porosity [-] | experiment duration [h:mm] | total outflow volume [L] | outflow lag [h:mm] | outflow duration [h:mm] |
|---|---|---|---|---|---|---|---|
| snow1 | 0.20 | 414 | 0.549 | 0:32 | - | - | - |
| snow2 | 0.20 | 438 | 0.522 | 1:10 | - | - | - |
| **firn1** | 0.16 | 459 | 0.499 | 2:07 | 4.3 | 0:38 | 1:29 |
| **firn2** | 0.16 | 451 | 0.508 | 2:39 | 2.1 | 0:40 | 1:59 |
| **firn3** | 0.17 | 600 | 0.345 | 1:21 | 5.1 | 0:24 | 0:58 |
| **firn4** | 0.28 | 538 | 0.417 | 1:54 | 2.1 | 1:06 | 0:48 |
| **firn5** | 0.19 | 506 | 0.447 | 1:47 | 3.2 | 1:02 | 0:45 |
| **firn6** | 0.13 | 574 | 0.374 | 1:30 | 3.7 | 0:48 | 0:43 |
| **snow3** | 0.18 | 406 | 0.557 | 1:31 | 2.2 | 0:57 | 0:34 |

**Table 2.** Measured and calculated parameters for the percolation experiments.

| experiment | initial density [kg m$^{-3}$] | final density [kg m$^{-3}$] | added mass [kg] | hydraulic conductivity [10$^{-5}$ m s$^{-1}$] | permeability (Darcy-based) [10$^{-10}$ m$^2$] | permeability (Calonne) [10$^{-10}$ m$^2$] |
|---|---|---|---|---|---|---|
| firn1 | 459 | 496 | 3.0 | 195 | 3.57 | 96.50 |
| firn2 | 451 | 491 | 3.3 | 47 | 0.87 | 107.58 |
| firn3 | 600 | 649 | 4.0 | 356 | 6.50 | 16.12 |
| firn4 | 538 | 591 | 7.6 | 224 | 4.09 | 37.02 |
| firn5 | 506 | 569 | 4.8 | 304 | 5.55 | 52.82 |
| firn6 | 574 | 660 | 5.2 | 262 | 4.79 | 22.49 |
| snow3 | 406 | 478 | 5.5 | 285 | 5.20 | 47.50 |

0.25 ± 0.091 m hr$^{-1}$. Outflow started before the entire firn block reached 0°C (Fig. 6). As the experiments progressed, firn temperature continued to increase. Once all sensors inside the firn block showed a temperature of 0°C, consistent outflow had already begun.

Piping and preferential flow was clearly visible in many of the experiments (Fig. 7). Note that in Fig. 7a and 7b sections of firn blocks are shown after water percolation which means that higher concentrations of dye highlight icy layers and preferential flow fingers, whereas Fig. 7c displays a section of a snow sample in which dye accumulates in finer-grained layers due to capillary forces. The fact that firn temperatures were locally still sub-zero but outflow was already taking place (Fig. 6), is further evidence that preferential flow occurred.



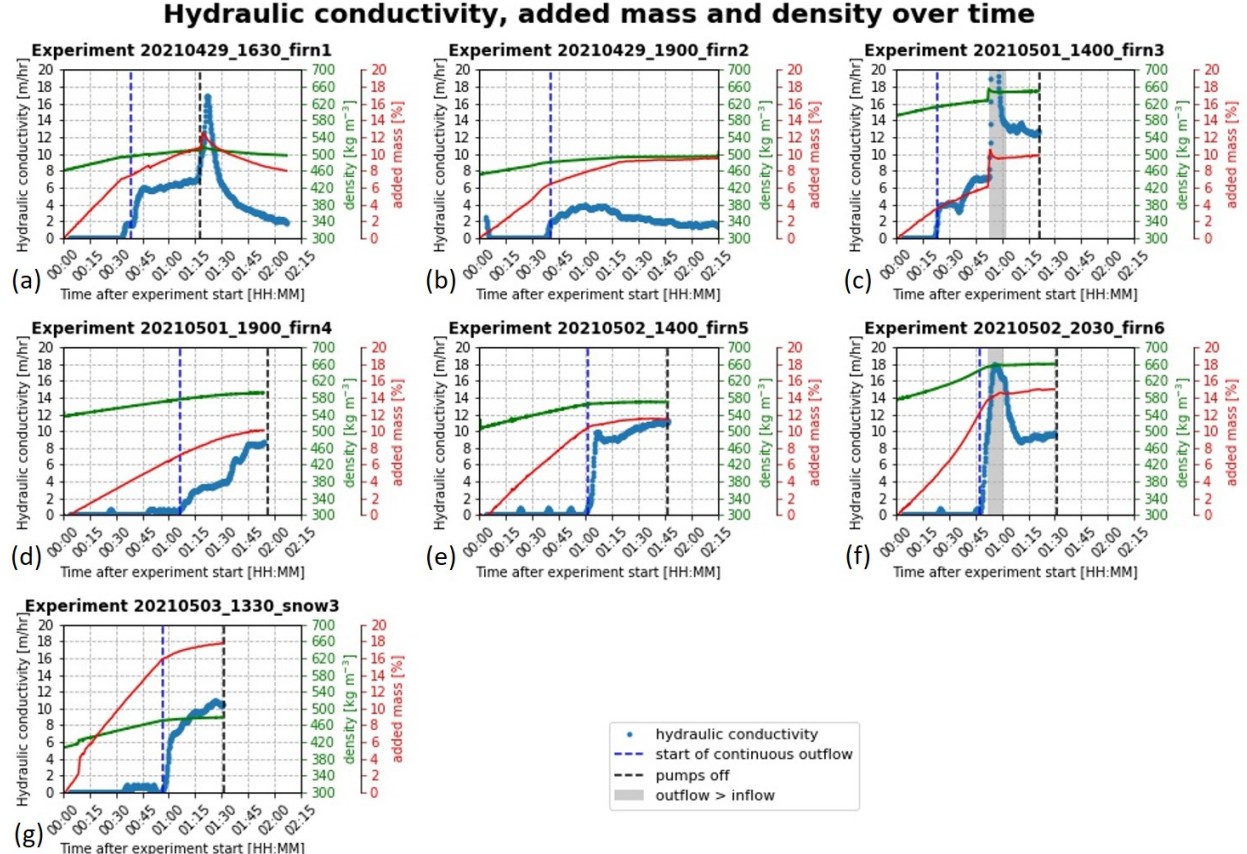

**Figure 5.** Hydraulic conductivity, added mass and density over time. In blue dots the calculated hydraulic conductivity, in red the firn sample mass as a percentage of its mass pre-experiment, and in green the density over time. The blue dashed line indicates start time of continuous outflow, the black dashed line shows the time at which water supply was stopped. Grey shading shows where outflow>inflow.

During experiment *firn1*, the movement motor that displaced the sprinkling head stopped working. When trying to move the sprinkling head manually, it fully drained instantaneously. Since this probably only occurred after the firn block had completely reached 0°C (Fig. 6), all sensors were left in the firn sample to continue measuring and the firn block was let to drain fully. Due to the lack of water supply, outflow quickly diminished (Fig. 5). Experiment *firn2* was carried out with a stationary sprinkling head, and as a result water droplets created narrow but deep holes within the firn block. During experiment *firn3*, almost 30

minutes after continuous outflow had started but well before the block was isothermal, an inadvertent 0.5–1 minute long peak of water inflow caused nearly instantaneous warming of about 5°C (Fig. 6). About 2 minutes thereafter, this inflow peak was also clearly visible in the outflow curve (Fig. 5). In experiment *firn4*, with the firn sample containing the thickest and ostensibly fully continuous ice lens, water was observed to flow around this ice lens on the side of the firn block. We measured the volume




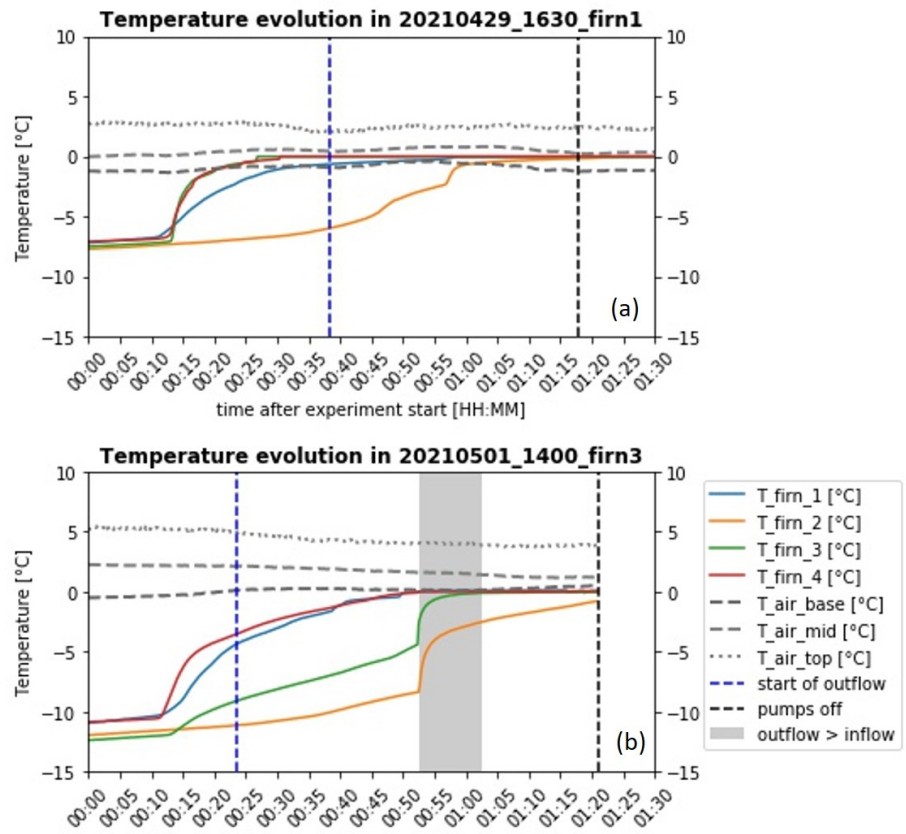

**Figure 6.** Air- and firn temperature over time for experiments *firn1* and *firn3*. The blue dashed line indicates start time of continuous outflow, grey shading shows where outflow>inflow. The black dashed line shows the time at which the experiment was stopped (when pumps were turned off). Coloured lines show temperature evolution at 4 locations within the firn block, ∼1 cm above its base. Grey dashed and dotted lines represent air temperature next to ROSA.

of water bypassing the ice lens, by repeatedly weighing the amount of water absorbed by tissues pressed against the ice lens for a fixed period of time, making sure that no water was sucked into the tissue from within the firn above and below the ice lens by capillary forces. On average, 35 out of the total 42 ml min⁻¹ water outflow was found to not flow through but around the ice lens. This would mean that about 15% of the total measured outflow was still percolating through the ice lens within the firn sample. Given the uncertainty of the method, however, it is unclear whether any water actually percolated through the ice.





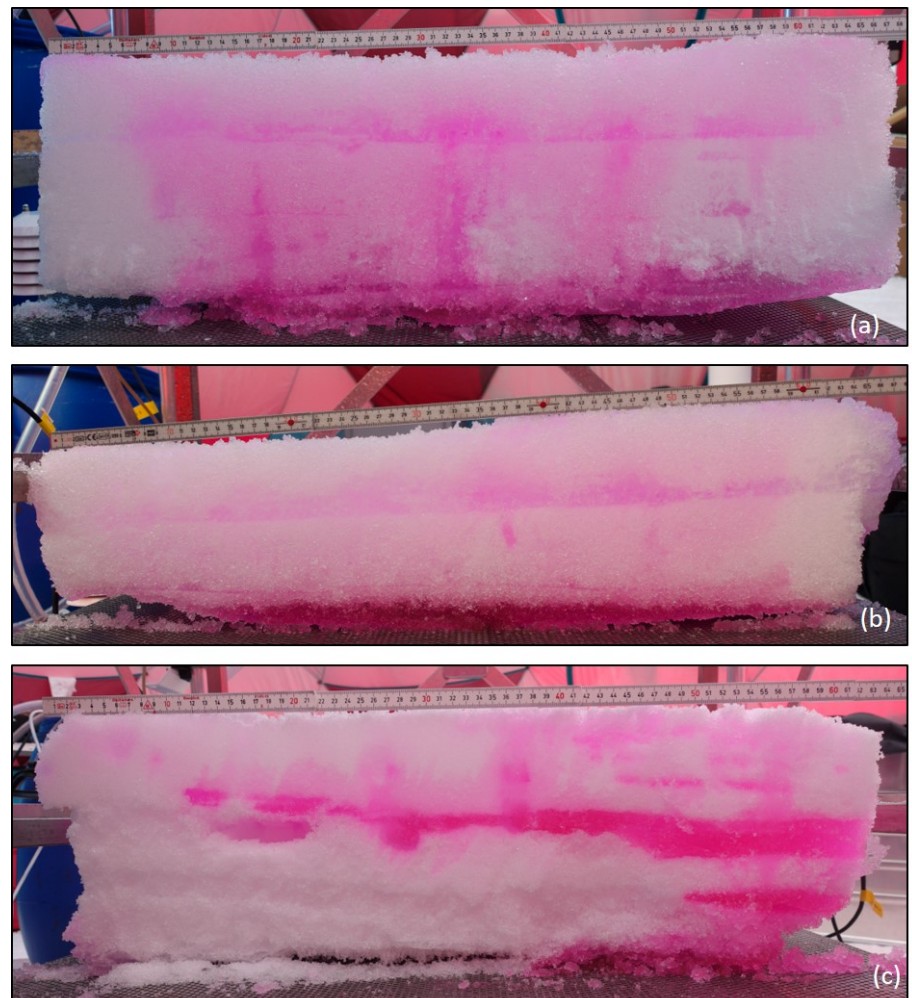

**Figure 7.** Three examples of firn and snow block sections after water percolation showing preferential flow paths and structural heterogeneity. Figures (a) and (b) are firn samples (*firn3, firn6*) where higher concentrations of dyed water highlight icy layers and preferential flow fingers, whereas figure (c), displaying the sample after experiment *snow3*, shows that in mature snow dye accumulates in finer-grained layers due to capillary forces.

## 5   Lateral meltwater flow in slush

### 5.1   Methods

During the summer field campaign in July-August 2020, we measured lateral flow velocity around FS2 (see Fig. 2). We drilled a borehole transect to investigate parameters that influence meltwater travelling through a slush matrix over the near-surface ice slab (Fig. 8a).



Along the transect, we drilled shallow cores and dug a number of snow pits to the top of the ice slab, logged these for grain
size and wetness, and noted the presence of ice lenses and layers. Depth of the ice slab below the snow surface was measured
either using a tape measure/folding ruler, or by employing an avalanche probe. To determine water table height we applied the
steel-tape method (Cunningham and Schalk, 2011), using a folding ruler and a chalk marker pen. Water table height here is
defined as the thickness of the water column on top of the ice slab, after the water level has nearly instantaneously equilibrated
following drilling of the borehole and snow removal from the hole.

Porosity measurements were made using a measuring cylinder which was inserted into fully water-saturated matrix and then
carefully extracted not to lose any liquid or slush matrix. Subsequently the weight of the filled cylinder was determined, liquid
water was poured off and the cylinder's weight with now empty matrix was determined. We measured the volume and weight
of water poured off as a cross-check, and using these measurements calculated slush matrix porosity following Eq. 2, taking a
value of 917 kg m$^{-3}$ for the ice density $\rho_i$ at 0°C.

To measure lateral meltwater flow rates, we used two different methods: salt dilution experiments and dye tracing. For the
salt dilution experiments, Darcy's law allows for calculating the flow velocity based on the concentration decay of the used
tracer (Freeze and Cherry, 1979):

$$q = \frac{-\pi r}{2t\alpha} \ln \frac{C}{C_0} \tag{11}$$

where $q$ is the meltwater flow rate [m s$^{-1}$], $r$ is the borehole radius [m], $t$ is time [s], $\alpha$ is the "drainage coefficient" required
to correct the flow velocity for borehole effects, commonly taken to be two (Pitrak et al., 2007), and $C/C_0$ is the relative
concentration at a given time (Miller et al., 2018). After measuring the water depth and determining the background conduc-
tivity of the meltwater in a borehole with a Hanna Instruments HI98195 multi-parameter sensor, a dilute salt-water solution
(10 g L$^{-1}$ kitchen salt) was injected into the borehole. The subsequent conductivity decay was measured to determine the lateral
meltwater flow velocity over the ice slab (see Fig. 8b for a schematic overview of the measurement set-up).

The dye tracer experiments were carried out to visually determine lateral meltwater flow velocity and confirm unidirectional
flow. We injected liquid Rhodamine WT (RWT) into the meltwater on top of the ice slab, and at multiple locations visually
determined the timing of first occurrence of the tracer in thin trenches. These were dug before the start of each experiment,
to avoid disturbing the overlying snowpack and meltwater flow as much as possible. After identifying the dyed water in all
trenches, we removed the overlying snowpack to study the complete flowpath.

All meltwater flow velocity measurements, as well as the slush property measurements, were carried out at locations where
there was no visible water at the surface, i.e. where meltwater had not yet fully saturated the snow and firn on top of the ice
slab (Fig. 2b).

## 5.2 Results

### 5.2.1 Slush matrix properties

During the summer 2020 field campaign, a total of 27 slush samples from along the borehole transect were collected. All
samples consisted of rounded, sometimes clustered ice grains and water (MFsl and MFcl according to the classification for





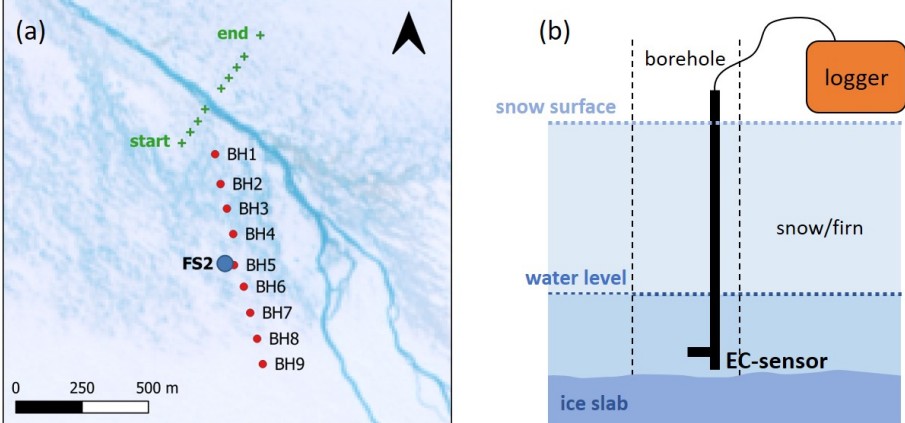

**Figure 8.** (a) Location of boreholes (red points) and river bed transect (green crosses) on the GrIS during the summer 2020 field campaign. The background image is a Sentinel 2 true color composite from 12.08.2019, around the time of maximum melt extent that year. (b) Schematic representation of the measurement set-up during salt dilution experiments.

seasonal snow by Fierz et al., 2009) of about 1.5–3 mm in diameter for individual slush grains and up to multiple cm's for the clusters (Fig. 9a). We found little lateral variation in slush properties, although the overlying dry(er) snow showed some variation in grain size and in particular hardness. In slush located below the water table we could not identify any visual

variation in matrix properties.

Total porosity of the slush samples was determined between 18 and 67%, with a mean of 41% and a standard deviation of 10%. Some residual water remained in the firn samples after porosity measurements (Fig. 9b). Actual porosity therefore is likely somewhat higher than the measured values, ranging between 23 and 72% with a mean of 45 ± 10% (assuming a residual LWC of 4%, see section 3).

**5.2.2 Lateral meltwater flow velocity measurements**

A total of 85 salt dilution experiments were carried out along the shallow borehole transect, resulting in an average flow velocity of ~7.0 ± 10.0 m hr$^{-1}$ (= 192 ± 277·10$^{-5}$ m s$^{-1}$), with minimum and maximum trusted measured velocities between 1.3–14.2 m hr$^{-1}$. Trusted velocities do not include very low and high measured velocity values that are potentially due to measurement errors related to the EC-probe's high sensitivity to sensor positioning. Table 3 shows a summary of the salt

dilution measurements, displaying average flow velocities measured on various days throughout the melt season. An example curve of the salt concentration decay measured during one of the experiments can be found in Appendix A (Fig. A1). BH1 is closest to the main river system and BH9 furthest away (Fig. 8), but neither in terms of flow velocity nor in terms of water table height is there a relationship with distance to the main drainage channel. No clear temporal trends are visible from these results either, nor is there a significant correlation between lateral meltwater flow velocity and water table height (Fig. 10, P-value of

295 0.20).



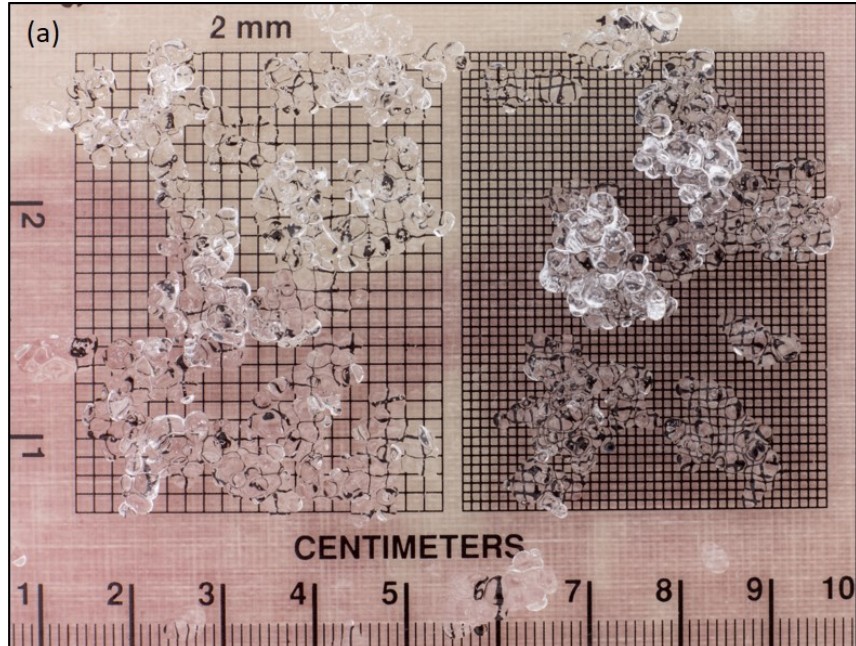
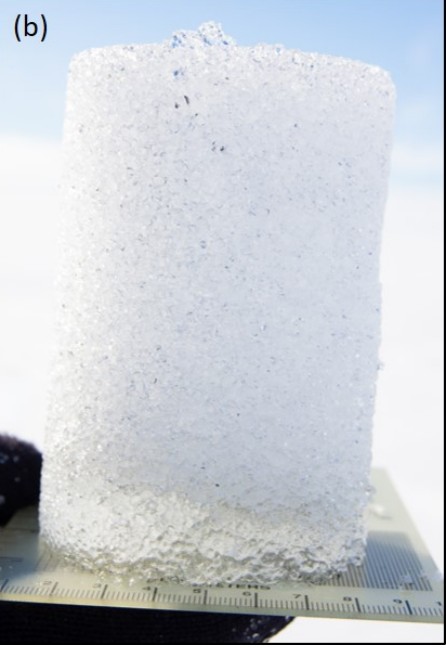

**Figure 9.** (a) Sample of slush matrix showing relatively homogeneous 2 mm-large rounded melt forms. (b) Slush sample showing water retention due to capillary forces (residual water)

**Table 3.** Overview of average flow velocities measured on various dates in all the boreholes along the transect, in m hr$^{-1}$. Values represent average velocities measured on a specific date: in most cases 3 experiments were carried out per borehole per day but only the average of these velocities is displayed here. Values denoted with an asterisk (*) are likely overestimations of the actual flow velocity due to sensor misorientation.

| Date | BH1 | BH2 | BH3 | BH4 | BH5 | BH6 | BH7 | BH8 | BH9 |
|---|---|---|---|---|---|---|---|---|---|
| 23.07.2020 | 1.51 | | 0.87 | | 4.60 | | | 3.46 | |
| 26.07.2020 | 6.27 | 7.25 | 4.36 | 9.19 | 31.66* | 6.50 | | 38.72* | |
| 27.07.2020 | | | 6.79 | | 2.57 | | | 14.94 | |
| 28.07.2020 | 3.99 | | | | 2.77 | | | | |
| 29.07.2020 | 2.49 | | 2.94 | | 1.12 | 1.47 | | 6.87 | |
| 30.07.2020 | 0.36 | | 1.65 | | 1.83 | 3.35 | 5.25 | 3.77 | |
| 31.07.2020 | | | | | | | | | 6.13 |
| 01.08.2020 | | | 5.10 | | | | | 5.19 | |

We undertook several dye tracing experiments, yielding an average flow velocity of ~7.0 m hr$^{-1}$ in a total range of 3.5–15.1 m hr$^{-1}$. Dye tracing revealed that meltwater flow over the ice slab is clearly directional (Fig. 11). Differences in RWT dye





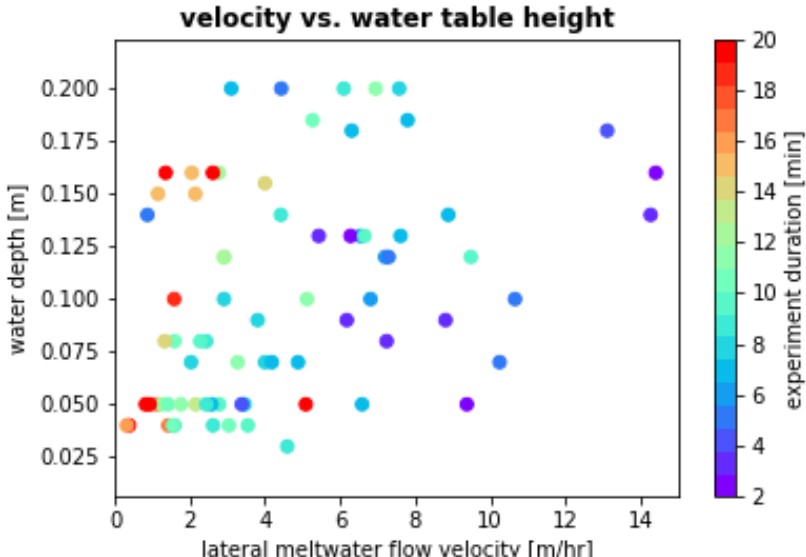

**Figure 10.** Flow velocity vs. measured water table height in the salt dilution experiments. Colors indicate experiment duration.

concentration (Fig. 11d) show that small localised meltwater ponds are present on top of the ice slab due to small-scale surface roughness.

An initial warm and sunny period was followed by decreasing temperatures and cloudy weather with precipitation, initially some rain followed by intense snowfall. Figure 12 shows the air temperature and cloud cover measured at the KAN_U weather station for the 2020 melt season. The KAN_U weather station is part of the Programme for Monitoring of the Greenland Ice Sheet (PROMICE; Ahlstrøm et al., 2008). Despite the decrease in temperature and nearly continuous full cloud cover from halfway through the campaign onwards, liquid water remained present at the ice sheet surface throughout the fieldwork period

and for even longer on top of the near-surface ice slab (see also Fig. 2b).

### 5.3 Theoretical determination of lateral meltwater flow velocity

We calculated lateral flow velocities for meltwater flowing through the slush following Darcy's law (Eq. 8). To obtain permeability, we used both the Kozeny-Carman equation (Eq. 9) and Calonne's parametrisation (Eq. 10) for perfectly spherical grains in a matrix with a porosity of 0.25 and 0.50, based on our slush property measurements. Equivalent sphere radius was

set to half the observed grain size of the matrix, since snow/ice particles in the slush were near-perfect spheres. We set the ice slab slope equal to the local ice sheet surface slope along flow lines of supraglacial streams visible on satellite imagery (around 5 m elevation difference per kilometer, which equals a slope of ∼0.30° based on ArcticDEM V1; Porter et al., 2018).

Resulting lateral flow velocities range from 0.073 to 1.31 m hr[-1] for the permeabilities obtained using the Kozeny-Carman equation, and between 0.052 and 0.96 m hr[-1] for permeabilities according to Calonne's parametrisation. These are up to 3 orders





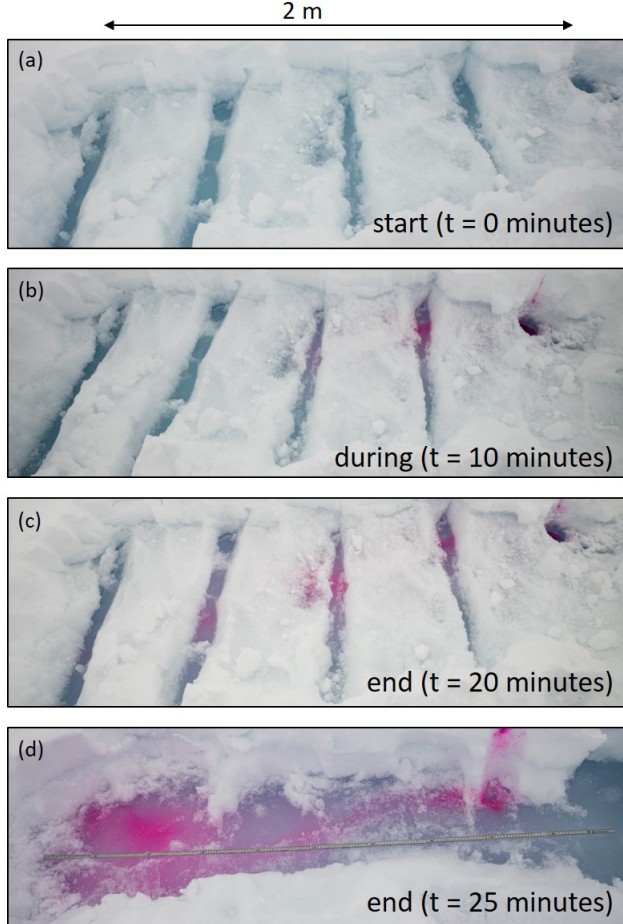

**Figure 11.** Results from a RWT dye tracing experiment, showing the monitored site at the time of dye injection (a), halfway (b) and at the end of the experiment before (c) and after (d) full exposure of the flow path. Rhodamine WT was injected in the borehole (in the top right corner of the image, located upstream). Note that on plot (d) the snow was also excavated uphill from the insertion point.

of magnitude smaller than the flow velocities measured in the tracer experiments. Using Darcy's law to back-calculate values of permeability for measured flow velocities during summer (1.3–15.1 m hr$^{-1}$) results in values between $3.55 \cdot 10^{-8}$ and $1.53 \cdot 10^{-7}$ m$^2$ for an ice slab slope of $\sim$0.30°. This is up to 3 orders of magnitude larger than the permeabilities calculated based on the slush matrix properties using either the Kozeny-Carman or Calonne parametrisation.



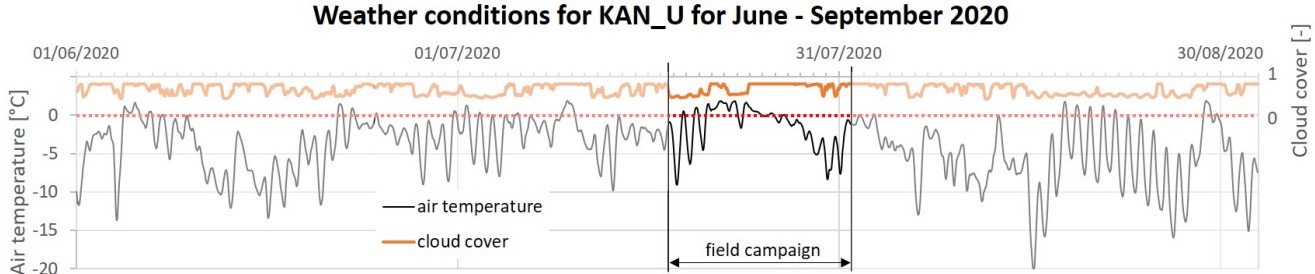

**Figure 12.** Air temperature and cloud cover at KAN_U weather station from June 1st to September 1st, 2020 (source: PROMICE)

## 6 Discussion

### 6.1 Meltwater flow velocities through snow and firn

Flow velocities through snow and firn were determined in two ways: directly, by measuring lateral flow over ice slab, and by calculating the hydraulic conductivity from ROSA-data using Darcy's law. Here we compare these two parameters as firn hydrological properties, although it should be noted that they are not exactly equal. The measured lateral flow velocities are not only a function of the firn hydraulic conductivity, but also governed by other external factors such as ice slab slope, and therefore not purely material properties of the firn. Furthermore, the calculated hydraulic conductivities are based on meltwater percolation through preferential flow fingers. It is uncertain to what extent the preferential flow paths are saturated, but it is clear that water saturation was highly variable in the meltwater percolation experiments. Derived hydraulic conductivities therefore likely underestimate the saturated hydraulic conductivity.

Figure 13 shows a comparison of our measured flow velocities to other studies measuring flow speeds through snow, firn and firn aquifers. The velocities were determined using different methods, including slug tests (Kruseman et al., 1994), snowshed lysimeters, measuring lagtimes of meltwater poured on the snow surface reaching a certain depth, and using dielectric sensors detecting changes in LWC. The percolation velocities do not all capture the same process: in some cases the values show the percolation velocity of water through preferential flow fingers, whereas velocities resulting from slug tests measure hydraulic conductivity in fully saturated snow or firn. Furthermore, some measurements relate to vertical flow, whereas others present values for lateral flow velocities. Thus, not all velocity ranges in Fig. 13 can be compared directly. Measured velocities presented in this paper are relatively high compared to existing values, but they overlap with measurements by e.g. Gerdel (1954); Kattelmann (1987); Fountain (1989) and Miller et al. (2017). Note that the vertical percolation values for ROSA are a combination of the velocities for unsaturated meltwater percolation and saturated flow through preferential flow fingers. Measured Darcy velocities (hydraulic conductivity) for saturated water percolation through the firn are in the same order of magnitude as the observed lateral matrix flow of meltwater on top of near-surface ice slabs. The lower flow velocities for unsatured vertical meltwater percolation show more overlap with some of the existing measurements, but are still relatively high.



**Figure 13.** Flow velocities through snow and firn as measured in this study, compared to other values published in literature. Velocities measured in firn are represented in **bold**, 'normal' labels are for values measured in snow.

### 6.1.1 Vertical percolation experiments

Meltwater percolation in the ROSA-experiments predominantly occurs through flow fingers, and water ponds on (relative) permeability barriers such as ice lenses or grain size contrasts. The start of continuous outflow does not mean that firn samples

were fully saturated: density continued increasing once outflow had started, and temperature sensors indicated heterogeneous warming of the block. Mass gain and hence densification occurred at different rates before and after the start of continuous outflow. This is likely due to an initial increase in sample water saturation, until sufficient preferential flow paths have formed and water saturation is locally high enough for outflow to occur. There is no obvious correlation between firn sample density and outflow lagtime.

Hydraulic conductivity of firn as measured for unsaturated flow varied between 0.17 and 0.44 m hr$^{-1}$ with an average of 0.25 $\pm$ 0.091 m hr$^{-1}$. In the final part of the experiments, when preferential flow paths had sufficiently developed, hydraulic conductivities ranged from 1.71 to 12.80 m hr$^{-1}$ with an average of 8.60 $\pm$ 3.58 m hr$^{-1}$. The order of magnitude difference between





hydraulic conductivity of unsaturated- and 'saturated' meltwater percolation clearly shows the efficiency of preferential flow paths. It is unclear whether formation of preferential flow paths is still ongoing or has completely finalized by the end of the

percolation experiments, as it appears that firn blocks were not yet fully in equilibrium state (albeit close – the slope of the hydraulic conductivity plots in Fig. 5 is almost but not completely zero). Hirashima et al. (2019) conducted computer simulations to investigate the transition from preferential to matrix flow, and showed that preferential flow paths likely continue migrating through the snowpack over time because of wet snow metamorphism. This suggests that preferential flow path development is not a finite process.

Even though ice lenses of up to several cms thick were present they never impeded percolation fully, with the exception of the 3–5 cm thick ice lens present in the sample used for *firn4*. According to the measurements of flow around the ice lens during this experiment, a maximum of 15% of the supplied water percolated through the firn block. Observations of dyed water presence in the center of the block after the experiment are inconclusive – it is unclear whether the minor amounts of pink dye present should be attributed to cutting artefacts from after the experiment or whether its presence was a result of actual flow

through the ice lens.

Firn permeabilities calculated using ROSA data range between $0.87 \cdot 10^{-10}$ and $6.50 \cdot 10^{-10}$ m$^2$, very similar to the lower estimates for the saturated slush permeabilities when using the parametrization by Calonne. On average, the permeability values resulting from the approximation by Calonne et al. (2012) are at least an order of magnitude larger than the permeabilities as calculated using Darcy's law. This could be related to the fact that the water saturation in the observed preferential flow paths is

unknown. Calculating values for unsaturated permeability is possible in theory, but this would require more detailed knowledge on the firn grain- and pore size distribution, and hence add significant measurement challenges.

Given the relatively large size of the firn blocks used in the ROSA-experiments, we assume that the measured hydrological properties of icy firn are representative more generically. However, we observed that firn properties (i.e. ice content, or water saturation at the end of all experiments) are highly variable within individual samples. Based on the firn cores that we drilled

at FS4, very close to the location of firn sample collection, the ice content of the samples used for the percolation experiments is significantly below the average ice content in the upper 10 m of the firn at this location (54% in the FS4-firn core, whereas maximum estimated ice content in the firn blocks was approximately 20%). This is a significant but necessary shortcoming of our measurements: the icier the firn, the larger samples would have to be to sufficiently accommodate lateral flow to adequately represent the percolation process. As far as we are aware, no observations exist that quantify the ratio between ice lens thickness

and -width. The lateral continuity of ice lenses, and hence the representativeness of the used samples for average firn properties, therefore remains uncertain. We observed no significant relationship between ice content and outflow lag time or hydraulic conductivity, which further emphasizes the uncertainty in spatial representativeness of the experimental results.

### 6.1.2   Lateral meltwater flow velocity

Measured lateral flow velocities of meltwater over the ice slab range from 1.3 to 15.1 m hr$^{-1}$. We note that there is good agree-

ment between velocities resulting from the salt dilution- and dye tracing experiments. The RWT measurements furthermore validate the results of the salt dilution experiments as velocities of directional flow and not just the speed of omnidirectional





tracer dispersion in a larger water body. Local ponding of laterally meltwater flowing occurs (see Fig. 11d), so measured flow velocities might be a combination of relatively fast directional flow and temporary local meltwater storage. This could also explain the considerable variability in the velocities resulting from the salt dilution experiments.

Calculated lateral flow velocities using Darcy's law, slush matrix properties and ice sheet surface slope result in significant underestimates compared to the measured meltwater flow velocities over the ice slab during the summer campaign: the calculated values are 1 to 3 orders of magnitude smaller than the measured velocities (0.073–1.31 m hr$^{-1}$ and 0.052–0.96 m hr$^{-1}$ using the Kozeny-Carman and Calonne's parametrisation for permeability, respectively, vs. 1.3–15.1 m hr$^{-1}$ as measured in the tracer experiments). Similarly, back-calculating permeability values for the slush matrix using observed meltwater flow

velocities leads to significant overestimation of permeabilities when compared to the results of commonly-used parametrisations by Calonne or Kozeny-Carman: $2.39 \cdot 10^{-7}$–$5.54 \cdot 10^{-8}$ m$^2$ according to measured velocities vs. $3.16 \cdot 10^{-10}$–$3.73 \cdot 10^{-8}$ m$^2$ and $7.41 \cdot 10^{-10}$–$1.33 \cdot 10^{-8}$ m$^2$ according to the Calonne and Kozeny-Carman parametrisations. Permeabilities calculated using Calonne's parametrisation and the Kozeny-Carman approximation result in very similar values, which is to be expected given the near-perfect sphericity of the ice grains in the slush matrix.

Hydraulic head variations between individual boreholes throughout the field work period were calculated to see if the assumption that the regional surface slope is the main driver for lateral meltwater flow is correct. This method, using water table height differences between individual boreholes to calculate flow velocities, also resulted in significant underestimation of meltwater flow rates when compared to the observed values.

## 6.2    Slush matrix properties & water table variation

Porosity of the slush was on average 41%, ranging from 18–67%, with a clear residual water saturation due to capillary forces. The lack of variation in slush matrix properties within the water column is likely due to the thermodynamic equilibration process between snow and ice at this high liquid water content.

    Local-scale ice slab topography has a significant impact on meltwater flow direction and to a lesser extent on flow velocity. There is no correlation between water table height and flow velocity, nor is there a link between distance to the main supraglacial

drainage channel. We therefore conclude that flow direction is principally governed by the regional ice sheet and -slab surface slope, but since this is very gentle, second-order factors like local firn stratigraphic features and small-scale ice slab surface undulations also affect meltwater flow direction. In some cases, the ice slab surface undulations that we found were surprisingly large (up to 50 cm within 3 m distance). However, we noticed that flow was present whenever the water was deep enough to do salt dilution experiments, i.e. roughly 5 cm. Since we furthermore found no correlation between water depth and flow speed, we

conclude that for water depths >5 cm, the water always finds a way around local undulations of the ice slab. Surface irregularity might change flow direction locally, but there is no evidence that this fully impedes meltwater flow.

    During the summer 2020 campaign, we observed that there was no clear link between weather conditions and water table height on top of the ice sheet. Changes in water depth only occured with a clear delay after changing meteorological conditions: when the weather turned colder and more cloudy there was an obvious time lag before any significant decrease of the water

table could be seen.





## 6.3    The slush- vs. the runoff limit

We show that runoff was occurring through the subsurface around FS2, even in the absence of visible slush. Runoff below the ice sheet surface continued for days after surface melting had stopped, resulting in break-throughs of the water table onto the surface that we saw in the field and could identify in visible satellite imagery (Fig. 2b, lower panel) The high lateral flow

velocities that we measured, and the ample presence of liquid water during our field campaign, are strong indications that FS2 was below the runoff limit, i.e. within the runoff area, during summer 2020.

Based on the lateral flow velocities presented here, and the maximum number of days on which meltwater transport occurred in 2012, we estimate the maximum distance between the slush- and runoff limit at 4 km. However, we suggest that the real difference is likely to be less, as water flow is inhibited by local subsurface ponding and flow direction is significantly influenced

by small-scale ice slab topographic variations, and our calculations assume that surface inputs are constant.

The evolution of slush limit altitude throughout the melt season has been investigated based on remote sensing data and by degree-day modelling (Reeh, 1991; Greuell and Knap, 2000), but since *in situ* measurements made at the runoff limit do not exist as of yet, it is challenging to determine (what governs) the distance between the slush- and runoff limit. Even though we have quantified vertical and lateral meltwater flow velocities through snow and firn near the runoff limit in this paper, we lack

other essential data to further constrain and describe the hydrological system in the accumulation area of the SW Greenland Ice Sheet.

## 7    Conclusions

We carried out fieldwork on the southwestern Greenland Ice Sheet around the K-transect, both in the region where near-surface ice slabs are present and where the firn has not yet substantially been affected by ice slab formation. We present a novel dataset

of hydraulic conductivity measured in icy firn, and, to our knowledge, the first measurements of slush properties and lateral meltwater flow velocity through this slush matrix over the ice slab.

Firn hydraulic conductivity measured in percolation experiments, ranging between 1.71 and 12.80 m hr$^{-1}$, is in the same order of magnitude as the measured lateral meltwater flow velocities through a slush matrix on top of near-surface ice slabs (1.3–15.1 m hr$^{-1}$ with an average of 7 m hr$^{-1}$). Conversely, lateral meltwater flow velocity calculated using Darcy's law results

in flow velocities of only 0.020–2.38 m hr$^{-1}$ with an average of 0.22 m hr$^{-1}$ (mainly depending on the slush density), which is about an order of magnitude lower than the lateral flow velocities observed in the tracer experiments.

These measurements are a first step towards an integrated set of hydrological properties of firn on the SW Greenland Ice Sheet, we have not yet been able to link vertical and lateral meltwater flow directly. We still lack understanding of the processes which drive the transition from meltwater flow dominated by vertical percolation to laterally-directed flow which contributes to

ice sheet runoff. Our data do, however, provide evidence that the slush limit and the runoff limit are not necessarily colocated, since we show that laterally flowing meltwater can be present above the slush limit in the accumulation area on the SW Greenland Ice Sheet.



*Author contributions.* NC together with HM and AT designed the study and collected field data, NJ assisted with data collection and field-work logistics. NW contributed to intellectual content, RW and OR provided fundamental knowledge and advice on operating ROSA.

NC conducted data analysis and interpretation and prepared the manuscript with contributions from all co-authors.

*Competing interests.* The authors declare that they have no conflict of interest.

*Acknowledgements.* We thank Olivia Miller for advice on measurement strategies as well as fruitful discussions on firn hydrology. This work was funded by the European Research Council (ERC) under the European Union's Horizon 2020 research and innovation programme (grant agreement No. 818994 – CASSANDRA) and by a SPARK grant, funded by the Swiss National Science Foundation (SNSF), project No.

CRSK-2_190845.




## Appendix A

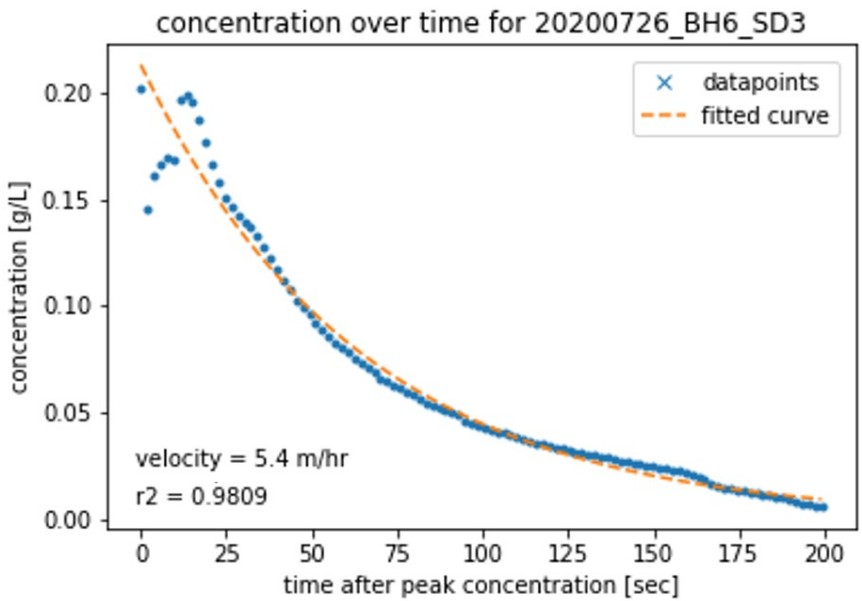

**Figure A1.** Example salt concentration decay curve showing fitted curve based on which meltwater flow velocity was calculated.

**Table A1.** Description of fieldsites where firn cores and shallow boreholes were drilled for tracer experiments, and where meltwater percolation experiments were carried out.

| Site name | Latitude (°) | Longitude (°) | Elevation (m a.s.l.) | Measurements conducted and field season |
|---|---|---|---|---|
| FS2 | 66.98605 | -47.23809 | 1765 | Tracer experiments, firn property measurements, July-August 2020; firn coring, April 2021 |
| FS4 | 67.01044 | -46.81707 | 1894 | Firn coring, meltwater percolation experiments, April-May 2021 |
| FS5 | 67.01025 | -46.46525 | 1977 | Firn coring, May 2021 |





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
