# Peer review of "In situ measurements of meltwater flow through snow and firn in the accumulation zone of the SW Greenland Ice Sheet"

_EGUsphere, 2022_

## Referee Comment (RC1)

Review of the manuscript

In situ measurements of meltwater flow through snow and firn in the
accumulation zone of the SW Greenland Ice Sheet

by Nicole Clerx, Horst Machguth, Andrew Tedstone, Nicolas Jullien,
Nander Wever, Rolf Weingartner, and Ole Roessler

submitted for publication in The Cryosphere

Prepared by Sergey Marchenko, postdoctoral research fellow

University of Northern British Columbia and

Vancouver Island University, Canada

**General comments**

In the present manuscript authors address the questions of melt
water drainage from the areas above the equilibrium line of the
Greenland ice sheet. On the basis of results from two field
campaigns at 1700 – 2000 m asl on the K transect in SW Greenland
conclusions are made regarding such properties of the snow/firn/ice
as density, statigraphy, hydraulic properties. Most notably
quantification of both vertical and lateral water flow speeds
through snow and firn is reported from multiple field experiments.

Given the increasing melt rates reported from the Greenland ice
sheet and projected for the years to come, domain above the
equilibrium line undergoes rapid changes with the general pattern of
glacier zone migration upwards. Field evidences from the area,
particularly those of quantitative nature, are important for a
better understanding of the ongoing processes and are crucial for
their formal description in numerical models.

The manuscript is based on extensive field data, is well structured,
presentation is generally logical and consistent. Results can not be
said to report anything that was not observed earlier and do not
allow to make conceptually new generalizations. At the same time
this is a carefully prepared quantitative account on processes of
melt water infiltration and runoff from a very dynamic part of the
ice sheet and in my oppinion deserves to be made available to a
wider audience.

It appears to me, however, that the manuscript requires a number of
clarifications, additions and edits before publication.

**Specific comments**

| 1 | LL33... | I think that readers will appreciate a description of the observed and expected changes in glacier zones at the Gr ice sheet. The fact is that in a warming climate all zones move into higher altitudes. Cold firn is replaced by either warm firn or superimposed ice zone or even ablation zone depending on the regional climatic conditions. Otherwise it is not obvious how the slush conditions discussed here are related to the slush of the cited PFAs. |
|---|---|---|
| 2 | Ch. 3 | I think that here a few references are missing. First three equations, as well as the Darcy flow law are given without any citations. Smth. classical like:
- first snowpack paper: Bartelt, Lehning, 2002, https://doi.org/10.1016/S0165-232X(02)00074-5
- Jordan R, Albert M and Brun E (2008) Chapter: Physical processes within the snow cover and their parameterization. In Armstrong Richard and Brun Eric eds. Snow and Climate: Physical Processes, Surface Energy Exchange and Modeling. Cambridge: Cambridge University Press, pp. 12–69.
will be good here. |
| 3 | Eq. 4 | It appears to me that equation 4 might need corrections: density rho and grav. acc. g are missing in the rightmost component. |
| 4 | L104 | Since quantification of hydraulic conductivity is one of the focuses of the study it would make sense to spend some words explaining its physical meaning. The fact that it has the same units as the water flow velocity may be confusing and further justifies such a clarification. Smth. along the lines:
"property that describes the ease with which given liquid can move through porous media. It depends on ..." (from Wikipedia). |
| 5 | L139 | It would be good to specify what was the sprinkling pattern at the surface of snow. Did water come out as a single jet or it was sprayed in small drops in a 3D fan pattern. Also how many injectors were used and what was the distance between the injectors and the snow surface.
These are issues that potential readers may be wondering about in connection with the preferential flow patterns reported later. Particularly since "deep holes" are mentioned at line 147. The overall question is: "is it even possible that the observed preferential flow formation is caused by the spay pattern?" |
| 6 | L198 | Is it right that the discharge of water collected by the lysimeter and measured by the tipping bucket |

| | | during 15 min before the supply cut off was divided by the sample area? If yes, that'd be good to express that more explicitly. This technique is close to the constant head permeability test: http://www.geotechdata.info/geotest/constant-head-permeability-test. |
|---|---|---|
| 7 | L208, reference to eq. 10 | It is nowhere specified how the SSA appearing in equation 10 was quantified. |
| 8 | Eq. 11 and the method of velocity quantification through conductivity measurements | Time between what events?

Perhaps it is the time between the measured max conductivity and conductivity value C...

Then, since multiple time-vs-concentration data points can be chosen, one can get multiple estimates of q from the decay curve. This guess is confirmed by the figure A1 showing multiple dots.

First of all, readers can not be left guessing, more transparency in description of the applied routines is needed. What is also apparent from the figure is that the curve is far from being linear. That implies that choosing different values for time and concentration one may get vastly different values for q. A comment on that is crucial for reporting and interpreting the results. |
| 9 | Ch. 5.2.2. | It is likely that in the described setting the lateral variation in the potential energy is the driving force for water drainage through the snow matrix.

It would thus be valuable to present information about the slope and aspect of the surface terrain and of the ice layer on top of which water is drained, if there is any of such info available.

In L293 water table height is mentioned. What exactly is meant here? What serves as reference here? Is that height above the sea level (=geoid) or rather depth below the glacier surface or something else? |
| 10 | Ch. 6.1., paragraph 1 | This paragraph is very confusing. Readers are likely to be lost in the many methods and directions of the water flow in snow and firn.

The chapters presenting results above contain velocities:
- vertical, from ROSA experiments: 0.167 - 0.438 m / h
- lateral from salt experiments: 1.3 - 14.2 m / h
- lateral from dye tracing: 3.5 - 15.1 m / h.

On top of that come hydr. cond. quantifications, which are, of course, not the same thing, but they do have |

| | | the same units and in case of vertical water flow are the same as flow velocity, if I understand it right... A reader may be wondering: "in Ch. 4 at line 215-216 reported velocities are claimed to be derived "*using the lag time and sample height*". How is that related to hydr. cond-s?"

Discussion will benefit from more precise formulations and also from a more thorough and consistent description of the background theory, which highlights the comment to line 104. |
|----|----|----|
| 11 | Methods chapters and L 366 | As far as i understand the permeabilities assessed from Darcy flow law rely on the results of the infiltration experiments yielding the K values. At the same time the k values parameterized following Calonne et al. (2012) rely on the measured density and SSA values. It is, as a matter of fact, nowhere explained how the latter are constrained.

This makes an important difference between the two kinds of k values, that is not properly highlighted in the text. The k values coming from Darcy flow law and K are, in a way, based on a more solid empirical dataset, but assume the validity of the D. flow law for the conditions of the experiment.

The latter fact calls for a more thorough explanation of the D. fl. law: what assumptions are implied and in what cases is it commonly used and was shown to do a decent job. |
| 12 | Results on firn hydr. Permeability and water flow velocity from different methods, L390-400 | This paragraph starting at L390 is largely a reiteration of the statements in the results chapter. Some explanation is expected here. It is a big thing when results from one method are off from results coming from another method by 10-1000 times. So something is seriously wrong with the quantification of the lateral water transport rate using different approaches. Either in the measurements/calculations routines or in the assumptions assumed by the methods.

Here and also in the results chapter, getting to the same conclusion from different ends (comparing velocities or permeabilities) appears more as a double check that one does to validate routines. But these are intermediate results providing auxiliary information that is important but not necessarily relevant in a publication. Readers can assume that results are solid and not be bothered by double checking. |
| 13 | L428 | References are needed when speaking of the high melt rates and spatial extent in 2012. Also give the actual number of melt days and an indication to where in |

| | | space they were observed to happen. That will make the link between the above given velocities of lateral water transport of ca 7 m per h and the 4 km offset presented here more transparent.

An additional factor possibly delaying runoff may be saturation of the likely thicker snow and firn higher up by the melt water before is becomes equally mobile as in the estimates presented in this study (7 m per h). Likely at the early stages of melt water can't move equally fast. |
|---|---|---|
| 14 | Reference list | I am not sure if TD standards allow including "in prep" publications in the list of citations. There are two such references: Machguth et al., Tedstone et al. |

**Comments to figures**

**Figure 2**

- May be break the figure in 4 panels A, B, C, D?
- Give names to the axis on second panel, easting, northing
- Clarification seems to be needed for what the readers can see at what is now panel b. The upper (and earlier) image appears to have more bright blue spots than the lower (and later) one. Intuitive interpretation also confirmed by the name of the plotted property (ndwi) is that these bright blue spots is surficial water. Then one may be wondering why is there less water later on in the season. That's counterintuitive: as cumulative melt increases one expects to see more water at the surface. It could be good to clarify this, alongside with pointing to the fact that melt water is seen higher up in the terrain on the later image (L 70). Or is it simply clouds that block the surface in the lower reaches of the later image?

**Figure 5**

- symbology is clear but presentation is not consistent: Change the font color for the left vertical axis and label to blue to make it apparent that the hydr. cond values are to be read on the left. This will also allow to get rid of the first legend entry. Alternatively: all fonts in black, but two more entires in the legend.
- It may be possible to give more space to the data curves by reducing the vertical axis labels and titles: they are the same for all panels and the grid will likely keep the curves readable even after keeping the axis attributes only at the very left and very right of the figure.
- titles of the panels of this and other figures. I'd suggest to start with the name (e.g. firn4) and give it in italic font to

match the text style. Then give the date and time. Year can be skipped and given in the figure caption.
- in the firn2 experiment the "pump off" time marker is missing? Did it get lost on the way or there was something special about this experiment?

**Figure 6**

- the symbols for the lower and middle air temperature curves are not distinguishable;
- the order of the three different air temperature legend entries is counterintuitive;
- the order of the air and firn temperature legend entries is counterintuitive, i'd suggest to have air on top in the legend box.

**Figure 13**

- figures in publications usually have no titles. Their function is taken by the caption.
- i'd suggest to "repack" this figure and adopt a structure that is closer to a table: move the references to the right or left so that they make one more column along. The other two columns will be the method and the actual values.
- combine the info from Kattelman (1987) by bringing closer together the individual data "pieces": I do not see why Ambach et al 1978 and Vallon et al 1976 need to be wedged in between. This will also allow to get rid of another "dimension" of the figure – grey shading behind references from Kattelman (1987) – in the updated more "table-like" structure that info can be given in the "references" column.
- regarding the rows of the "table". The existing structure is logical: methods make the higher order subdivisions, then references can define the lower-level subdivisions with a few grouped by the Kattelman (1987) figure bracket or similar. The original values first reported by this study could be either fitted in this structure or presented as a stand-alone group to make it more obvious what the study's contribution. In either case i think  2 more lines can be presented in this figure: vertical infiltration rates derived at time delay before onset of runoff after the start of spraying divided by the sample thickness (0.17 - 0.44 m / h) and lateral flow speed from dye tracing experiments.

**Technical corrections** are to be found in the accompanying *.pdf file.

[revised manuscript text omitted]

---

## Referee Comment (RC2)

Review of:
*In situ measurements of meltwater flow through snow and firn in the accumulation zone of the SW Greenland Ice Sheet*
Nicole Clerx, Horst Machguth, Andrew Tedstone, Nicolas Jullien, Nander Wever, Rolf Weingartner, and Ole Roessler

**Summary:**
This paper presents in situ measurements of water flow through firn in Southwest Greenland. The authors leveraged two methods. The first used a portable lysimeter, which measured vertical percolation of water sprayed by a sprinkler through blocks of firn that had been excavated from a pit. The second method used salt dilution and dye tracing experiments to investigate lateral meltwater flow velocities.

This work is an important contribution to the glaciological community: the fate of surface meltwater on the ice sheets is not well constrained, but it is becoming increasingly important to understand surface meltwater processes as an increasing area of Greenland is subject to melt. The research presented by the authors is novel and an important step in filling that gap in our knowledge; in situ measurements of any firn/meltwater processes are exceedingly rare.

This paper will be a good contribution to *The Cryosphere* after several issues, which I outline below, are addressed.

**General comments:**
1. My general feeling about this paper is that I was very excited about the science that was done, but the paper did not provide adequate discussion (1) about why the results were what they were; (2) about the broader implications of the research; and (3) about assumptions and uncertainties.

An example is that the theoretical, modeled lateral flow rates were 3 orders of magnitude less than the observed lateral flow rates. This is a very large discrepancy, but the discussion section does not include discussion of why this large discrepancy might exist. I encourage the authors to work on the discussion section to add more discussion of how their results corroborate or challenge our current understanding of firn meltwater hydrology. It might be useful to provide a simple, qualitative description of how these flow processes operate on different spatial scales based on the results. To add to the implications, can your results add any perspective to our understanding of the fate of firn meltwater; e.g. on longer time scales, what percent of meltwater is running off? Can your results be extrapolated to non-ice slab regions, or are they spatially limited?

The paper does not include uncertainty or error analysis, which I would like to see in a field study like this. I realize it can be difficult to quantify uncertainty in work like this, but even a short section qualitatively describing the uncertainties would improve the paper. For example, does permeability change during percolation with ROSA? How will that affect your quantitative results?

2.  A central assumption around your analyses is that the flow is Darcian. It may be the case that this is a valid assumption, but I think it would be appropriate to include justification. Is preferential flow in fingers/pipes Darcian? Would you calculate a different hydraulic conductivity if you conder Richard's Equation vs. Darcy's law? Is there a point at which you expect your Darcian assumption to break down? Your abstract says that for the ROSA experiments, "flow predominantly occurring through preferential flow fingers". Is there a difference in the conductivity between the preferential flow fingers and the matrix-flow instances? Do your calculated hydraulic conductivities (Equation 7 and line 198) represent a 'bulk' conductivity (that might represent conductivity for matrix flow) or the conductivity in the preferential flow pipes?

I think that it would also be appropriate for the introduction to more information on the current state of the science of matrix and preferential flow in firn(matrix flow is not mentioned until section 6), and subsequent sections could include more detail on how your experiments/results fall into these regimes. For example, in the ROSA experiments you observed preferential flow. Was it preferential flow from the start of the experiments, or was it initially matrix flow that evolved to preferential flow? If so, is there a timescale or threshold of when it transitioned? Is all flow in the lateral experiment assumed to be matrix flow?

2. For the ROSA experiments, water was introduced via a sprayer. This sounds to me like it is more simulating a rain on snow event than surface meltwater production due to a surface energy budget excess. I think it would be useful to discuss any differences you might expect in your results if it was surface meltwater production, which I would expect would be more uniform across the surface. Are the rates of water you are spraying comparable to the rate of meltwater production on a warm sunny day in SW Greenland? On line 231 you state that peak water flow caused a fast, large temperature increase, which made me wonder: was the sprayed water at 0C? It is probably worth specifying that in you methods.

3. A structural comment: Consider adding a bit of text at the end of the intro describing the structure of the paper, i.e. outlining, to clarify that there are two distinct but related experiments. You mentioned in the introduction that there were two field seasons/two experiments, but as I read section 4 I kept wondering about the other experiment.

4. The slush vs runoff limit finding is mentioned in the abstract, which indicates that it is an important result/outcome of the work. However, the discussion of this is only briefly mentioned at the end seems disparate from the results. I think it would be useful to add a bit more about how this 4 km is calculated and more discussion about the implications, including what additional "essential data" are required.

**Specific comments:**
Line 89: It may be worth specifying what a 'ripe' snowpack is for TC readers not familiar with snow hydrology.

136: "systematic measurements of parameters which are required to determine the hydraulic conductivity and water retention capacity of icy firn": this is very vague. Can you specify how they are systematic, and what the parameters are?

165ish: I would like a bit more detail about the samples here. What are the dimensions of one sample? Are the firn samples taken from different depths in the pit, or side-by-side extractions, (which would allow you to understand spatial variability, perhaps)? Table 1 could include a column that states the depth interval that the firn came from in the pit.

170: Can you be more specific about what is on this checklist?

201: Is the densification just due to adding mass to the sample, or is there compaction (volume change) too? Also, what is the 'apparent rate of densification'? Is that different than the actual rate?

203-205: I found the description of steps here a bit hard to follow. I suggest putting the description into past tense and write as a narrative; e.g. the water flow started, went this long, we observed X, then this happened, etc.

220: Is there a difference between piping and preferential flow? If not just say preferential flow was visible. Also, this is the only instance in the paper in which you use 'piping'; otherwise you use 'fingers', which I think is an interchangeable term. I suggest sticking with a single term.

Figure 5 – I suggest coloring the hydraulic conductivity axis (ticks and label) to be blue (same as the dot color) to be consistent with "density" and "added mass" axes. For clarity, I would remove the date_time portion of the subfigure titles, which will make the figure titles consistent with the naming in Table 1 and 2.

242: What parameters? I think you say in the next paragraph, but as a reader my initial reaction is that this is vague. I suggest reworking the text a bit to avoid this.

244: This is vague: what is 'shallow'? How deep were the snow pits?

246-249: The method here is vague – can you briefly explain the steel-tape method? Are you implying that you remove snow from the hole after drilling the borehole? Doesn't drilling a borehole inherently remove snow?

275: I think it would help clarify the text if your methods above use the same language of 'slush matrix properties' – it took me a moment to realize that your 'slush matrix properties' described in 5.2.1 were just the properties you were describing in 2$^{nd}$ and 3$^{rd}$ paragraphs of section 5.1. Also – it might be useful for you to include a more formal definition of what you mean by slush matrix.

Figure 10: It would be useful to add a bit more description to the caption, describing what it is that the reader should take away from the figure. Admittedly, I am not sure what I should take from the figure – simply that these are uncorrelated? I am not sure this figure is actually needed in that case; I think it suffices to state in the text that you did not find correlation.

Section 5.2.2/6.1.2: The discussion of the large variation (a factor of 10) in observed flow velocities is not adequately discussed. Why is there this large variation? Is it just local storage? Snow pack properties?

313: Consider adding language like "modeled flow velocities" throughout the text to clearly differentiate between when you are calculating theoretical velocities from an equation and your measurements.

Figure 12: I am not sure that this figure is needed, or if you want to include it consider adding a meltwater flux calculation from a surface energy balance model.

335: Can you be a bit more specific about which measurements are comparable? I.e., are the previous measurements that your data agree with capturing the same process?

340: typo, unsaturated

372: vague sentence – what is relatively large? What is 'more generically'?

387: "laterally meltwater flowing" – do you mean laterally flowing meltwater?

395: this seems to be a restatement of earlier, but still no why
447: rewrite sentence – incomplete at this point.

---

## Author Comment (AC1)

**Response to reviewer 1 (Sergey Marchenko)**

**In situ measurements of meltwater flow through snow and firn in the accumulation zone of the SW Greenland Ice Sheet**

Nicole Clerx, Horst Machguth, Andrew Tedstone, Nicolas Jullien, Nander Wever, Rolf Weingartner, Ole Roessler

Dear Reviewer,

We would like to thank you for your thorough and constructive review of our paper, and all the suggestions of how to improve its quality. Below, we respond point by point to all comments, and state how we plan to incorporate them in a revised version of the paper. The responses (normal font style) to the reviewer's comments are written directly into the reviews (displayed in italic font style). Revised figures are also included in this document. Technical corrections and/or replies to comments and suggestions made in the accompanying .pdf-file will be incorporated in a revised version of the manuscript.

Nicole Clerx,

*Fribourg, June 21, 2022*

**1 General comments**

*In the present manuscript authors address the questions of melt water drainage from the areas above the equilibrium line of the Greenland ice sheet. On the basis of results from two field campaigns at 1700 - 2000 m asl on the K transect in SW Greenland conclusions are made regarding such properties of the snow/firn/ice as density, statigraphy, hydraulic properties. Most notably quantification of both vertical and lateral water flow speeds through snow and firn is reported from multiple field experiments.*

*Given the increasing melt rates reported from the Greenland ice sheet and projected for the years to come, domain above the equilibrium line undergoes rapid changes with the general pattern of glacier zone migration upwards. Field evidences from the area, particularly those of quantitative nature, are important for a better understanding of the ongoing processes and are crucial for their formal description in numerical models.*

*The manuscript is based on extensive field data, is well structured, presentation is generally logical and consistent. Results can not be said to report anything that was not observed earlier and do not allow to make conceptually new generalizations. At the same time this is a carefully prepared quantitative account on processes of melt water infiltration and runoff from a very dynamic part of the ice sheet and in my oppinion deserves to be made available to a wider audience.*

*It appears to me, however, that the manuscript requires a number of clarifications, additions and edits before publication.*

Thank your for your thorough review. We agree that clarifying certain sections and adding/editing some of the information provided would improve the manuscript.

**2 Specific comments**

*(1) LL33...: I think that readers will appreciate a description of the observed and expected changes in glacier zones at the Gr ice sheet. The fact is that in a warming climate all zones move into higher altitudes. Cold firn is replaced by either warm firn or superimposed ice zone or even ablation zone depending on the regional climatic conditions. Otherwise it is not obvious how the slush conditions discussed here are related to the slush of the cited PFAs.*

We will include a more thorough description of the glacier facies on the Greenland Ice Sheet in general and where this study is situated in terms of the described glacier facies (at the moment and potentially in the future).

———————

*(2) Ch. 3: I think that here a few references are missing. First three equations, as well as the Darcy flow law are given without any citations. Smth. classical like:*

- *first snowpack paper: Bartelt, Lehning, 2002, https://doi.org/10.1016/S0165-232X(02)00074-5*
- *Jordan R, Albert M and Brun E (2008): Chapter: Physical processes within the snow cover and their parameterization. In Armstrong Richard and Brun Eric eds. Snow and Climate: Physical processes, surface Energy Exchange and Modeling. Cambridge: Cambridge University Press, pp. 12-69.*

*will be good here.*

OK, we will include those citations in the revised manuscript.

———————

*(3) Eq. 4: It appears to me that equation 4 might need corrections: density rho and grav. acc. g are missing in the rightmost component.*

Indeed, thanks for spotting this error. Will be corrected.

————

*(4) L104: Since quantification of hydraulic conductivity is one of the focuses of the study it would make sense to spend some words explaining its physical meaning. The fact that it has the same units as the water flow velocity may be confusing and further justifies such a clarification. Smth. along the lines: property that describes the ease with which given liquid can move through porous media. It depends on ..." (from Wikipedia).*

Agreed. We will include a better definition of the hydraulic conductivity.

————

*(5) L139: It would be good to specify what was the sprinkling pattern at the surface of snow. Did water come out as a single jet or it was sprayed in small drops in a 3D fan pattern. Also how many injectors were used and what was the distance between the injectors and the snow surface. These are issues that potential readers may be wondering about in connection with the preferential flow patterns reported later. Particularly since "deep holes" are mentioned at line 147. The overall question is: "is it even possible that the observed preferential flow formation is caused by the spay pattern?"*

The sprinkling head on ROSA contains 84 outlets that each have a diameter of 3.5 mm, and are arranged in a diamond grid. Ensuring they all contributed to irrigation more or less equally was part of the pre-experiment checks that were always carried out before starting an experiment. The sprinkling head was located 1 m above the so-called 'dripping plate' on which all samples rested. The total distance between the injectors and the snow/firn surface was roughly 85 cm on average (depending on the thickness of the sample, the distance between the base of the sample and the injector head was constant at 1 m).

The deep holes mentioned at line 147 are specifically due to (consciously) not moving the sprinkling head laterally w.r.t. firn block. During all other experiments the sprinkling head was moved in 5 cm-increments every 3 minutes to prevent these holes from forming. Furthermore, in the first two trial experiments, which are not described in the manuscript, we tested whether 'free fall acceleration' of the water droplets due to the distance between the sprinkling head and the top of the samples would affect the flow paths. We found this not to be the case, since the dyed water spread out laterally over the surface of the snow/firn samples quicker than initial vertical flow of water (i.e. the surface would be pink before any signs of vertical meltwater percolation could be seen on the sample sides, as well as before any outflow occurring).

We will include a better description of the sprinkling head and 'water delivery' to the samples in the revised manuscript.

————

*(6) L198: Is it right that the discharge of water collected by the lysimeter and measured by the tipping bucket during 15 min before the supply cut off was divided by the sample area? If yes, that'd be good to express that more explicitly. This technique is close to the constant head permeability test: http://www.geotechdata.info/geotest/constant-head-test.*

Yes, the volume of discharged water was divided by the sample area. It is true that this method is close to the constant head permeability test, but not exactly the same. The constant head permeability test works when being able to put the sample in an almost closed-off container to allow for full wetting/saturation, which was not the case for the ROSA set-up. Another difficulty in applying this method to our experiments is

that, even though outflow at some point stabilises, densification still occurs, which means that the sample is not yet fully saturated. This was confirmed by visual evidence after all experiments, none of the samples were fully pink so there were always unsaturated patches left. This means that resulting permeability values calculated in this way would always be a (significant) underestimation of the 'true' saturated snow/firn permeability.

————————

*(7) L208, reference to eq. 10: It is nowhere specified how the SSA appearing in equation 10 was quantified.*

The SSA was quantified using $r_{es} = 3/(SSA \cdot \rho_i)$, assuming that $2 \cdot r_{es}$ equals the average grain size. We will include this in the revised manuscript.

————————

*(8) Eq. 11 and the method of velocity quantification through conductivity measurements: Time between what events?*

*Perhaps it is the time between the measured max conductivity and conductivity value C...*

*Then, since multiple time-vs-concentration data points can be chosen, one can get multiple estimates of q from the decay curve. This guess is confirmed by the figure A1 showing multiple dots.*

*First of all, readers can not be left guessing, more transparency in a description of the applied routines is needed. What is also apparent from the figure is that the curve is far from being linear. That implies that choosing different values for time and concentration one may get vastly different values for q. A comment on that is crucial for reporting and interpreting the results.*

Equation 11:

$$q = -\frac{\pi r}{2t_i \alpha} ln(\frac{C}{C_0})$$

The time $t_i$ used in Eq. 11 is the time after the tracer injection. Rewriting this equation gives:

$$ln(C) = -\frac{2\alpha q}{\pi r} t_i + ln(C_0)$$

The dependence of the logarithm of average interval tracer concentration on time (i.e. the gradient/slope of the linear regression of $ln(C)$ vs. time) is proportional to horizontal flow velocity:

$$q = -0.5 * 1 \cdot \alpha \cdot \pi \cdot r \cdot slope$$

This is better explained in Pitrak et al. (2007), and also in section 9.4 of Freeze and Cherry (1979).

Note that in the current version of the manuscript, we give the apparent velocity, which is not the same as the average linear velocity of meltwater flowing through the pores (Eq. 9.28 in Freeze and Cherry, 1979). We will improve the description of this method/calculation in the revised manuscript, and also give the average linear velocity instead of the measured apparent flow velocity.

————————

*(9) Ch. 5.2.2: It is likely that in the described setting the lateral variation in the potential energy is the driving force for water drainage through the snow matrix.*

*It would thus be valuable to present information about the slope and aspect of the surface terrain and of the ice layer on top of which water is drained, if there is any of such info available.*

*In L293 water table height is mentioned. What exactly is meant here? Is that height above the sea level (= geoid) or rather depth below the glacier surface or something else?*

We agree that variations in the total hydraulic head (i.e. potential energy) is the driving force for water drainage through the snow matrix. Due to the very large heterogeneity in ice slab surface and the scale of the measured water depths (both are in the order of centimeters) compared to the overall slope of the terrain (0.30°, i.e. a depth difference of 0.5 cm per meter distance), it seems likely that local heterogeneities/undulations at the surface of the ice slab are more important in locally determining the direction than the slope and aspect of the surface terrain. (On a larger scale, we believe that the overall surface slope controls the flow direction.)

We define water table height as the thickness of the water column on top of the ice slab, after the water level has nearly instantaneously equilibrated following drilling of the borehole and snow removal from the hole.

We will include more detail on this in the results and discussion-section of the improved manuscript.

————

*(10) Ch. 6.1., paragraph 1: This paragraph is very confusing. Readers are likely to be lost in the many mehtods and directions of the water flow in snow and firn.*

*The chapters presenting results above contain velocities:*

- *vertical, from ROSA experiments: 0.167 - 0.438 m / h*
- *lateral from salt experiments: 1.3 - 14.2 m / h*
- *lateral from dye tracing: 3.5 - 15.1 m / h.*

*On top of that come hydr. cond. quantifications, which are, of course, not the same thing, but they do have the same units and in case of vertical water flow are the same as flow velocity, if I understand it right... A reader may be wondering: "in Ch. 4 at line 215-216 reported velocities are claimed to be derived "using the lag time and sample height". How is that related to hydr. cond-s?"*

*Discussion will benefit from more precise formulations and also from a more thorough and consistent description of the background theory, which highlights the comment to line 104.*

Thanks for pointing out that (this part of) the discussion is not clear. We will improve the discussion section in the revised manuscript.

————

*(11) Methods chapters and L366: As far as i understand the permeabilities assessed from Darcy flow law rely on the results of the infiltration experiments yielding the K values. At the same time the k values parameterized following Calonne et al. (2012) rely on the measured density and SSA values. It is, as a matter of fact, nowhere explained how the latter are constrained.*

*This makes an important difference between the two kinds of k values, that is not properly highlighted in the text. The k values coming from Darcy flow law and K are, in a way, based on a more solid empirical dataset, but assume the validity of the D. flow law for the conditions of the experiment.*

*The latter fact calls for a more thorough explanation of the D. fl. law: what assumptions are implied and in what cases is it commonly used and was shown to do a decent job.*

We will include a clearer description of how we established (measured) density and SSA values. Also the assumptions and validity of Darcy's law will be expanded upon to

better explain the methods we used to calculate and compare hydraulic conductivity- and permeability values across the two datasets.
* * *
*(12) This paragraph starting at L390 is largely a reiteration of the statements in the results chapter. Some explanation is expected here. It is a big thing when results from one method are off from results coming from another method by 10-1000 times. So something is seriously wrong with the quantification of the lateral water transport rate using different approaches. Either in the measurements/calculation routines or in the assumptions assumed by the methods.*

*Here and also in the results chapter, getting to the same conclusion from different ends (comparing velocities or permeabilities) appears more as a double check that one does to validate routines. But these are intermediate results providing auxiliary infomration that is important but not necessarily relevant in a publication. Readers can assume that results are solid and not be bothered by double checking.*

Although we strongly agree that the mismatch in permeability values resulting from our measurements/calculations is a major concern, we would also like to mention that permeability ranges in other natural porous media can cover multiple orders of magnitudes and permeability therefore is generally represented on a logarithmic scale. Sandstone permeabilities, for example, range between $10^{-10}$ and $10^{-15}$ $m^2$. In that sense variations of 3 orders of magnitude would not be unexpected. Nevertheless, since both the Darcy-based calculation and the Calonne-parametrisation were applied to the same samples, this discrepancy should of course not be there. We will further investigate this matter and provide a better discussion of the differences in permeability values and calculation methods in the updated version of the manuscript.
* * *
*An additional factor possibly delaying runoff may be saturation of the likely thicker snow and firn higher up by the melt water before is becomes equally mobile as in the estimates presented in this study (7 m per h). Likely at the early stages of melt water can't move equally fast.*

Comparison between vertical percolation (through preferential flow fingers) and lateral flow velocities shows that velocities are comparable, but there likely indeed is a threshold for the minimum amount of generated melt water required before lateral flow starts (i.e. sufficient volume to overcome local undulations in ice slab surface and build up a substantial water column).

In future work we will investigate changes in flow behaviour with various amounts of melt and different scenarios for melt input timing.
* * *
*(14) Reference list: I am not sure if TD standards allow including "in prep" publications in the list of citations. There are two such references: Machguth et al., Tedstone et al.*

Thanks for pointing these out. The references will be updated or removed in the revised manuscript.

**3   Comments to figures**

**Figure 2**

- *May be break the figure in 4 panels A, B, C, D?*
- *Give names to the axis on second panel, easting, northing*
- *Clarification seems to be needed for what the readers can see at what is now panel b. The upper (and earlier) image appears to have more bright blue spots than the lower (and later) one. Intuitive interpretation also confirmed by the name of the plotted property (ndwi) is that these bright blue spots is surficial water. Then one may be wondering why is there less water later on in the melt season. That's counterintuitive: as cumulative melt increases one expects to see more water at the surface. It could be good to clarify this, alongside with pointing to the fact that melt water is seen higher up in the terrain on the later image (L 70). Or is it simply clouds that block the surface in the lower reaches of the later image?*

Please find below an updated version of the figure and its header, as would be incorporated in the revised manuscript.

[Figure]

**Figure 2:** (a) Overview map of Greenland, with the black star indicating the approximate field site location. (b) Map of the study area, showing the various sites on the Greenland Ice Sheet (FS2 for summer measurements, FS4 for spring data collection, both sites and FS5 for firn stratigraphy and KAN_U for meteorological data). Thin black lines represent elevation contours from the ArcticDEM modified to show elevation in m a.s.l. (Porter et al., 2018). The background image is a Sentinel-2 true color composite from 12.08.2019, around the time of peak melt that year. The dashed dark blue rectangle indicates the outline of the composites shown in panels c and d. (c) Sentinel-2 NDWI composite showing the liquid water presence (in bright blue) on the ice sheet surface on 22 July 2020. (d) Sentinel-2 NDWI composite showing the liquid water presence on the ice sheet surface, again in bright blue, on 3 August 2020. Note that the surface meltwater in the lower areas is masked by the presence of clouds. For (c) and (d): the black star indicates the location of field site FS2, the NDWI composites have not been corrected for cloud artefacts. For (b), (c) and (d): source: sentinelhub Playground.

**Figure 5**

- *symbology is clear but presentation is not consistent: Change the font color for the left vertical axis and label to blue to make it apparent that the hydr. cond values are to be read on the left. This will also allow to get rid of the first legend entry. Alternatively: all fonts in black, but two more entires in the legend.*
- *It may be possible to give more space to the data curves by reducing the vertical axis labels and titles: they are the same for all panels and the grid will likely keep the curvves readable even after keeping the axis attributes only at the very left and right of the figure.*
- *titles of the panels of this and other figures. I'd suggest to start with the name (e.g. firn4) and give it in italic font to match the text style. Then give the date and time. Year can be skipped and given in the figure caption.*
- *in the firn2 experiment the "pump off" time marker is missing? Did it get lost on the way or there was something special about this experiment?*

Thanks for your suggestions, we have incorporated them in the updated figure & caption below.

The "pumps off" marker for *firn2* is missing because we left this experiment running for longer than is shown in the figure. The sprinkling head was not moved during this experiment, and this leads us to believe that the last part (i.e. beyond 02:15) is not representative anymore: at this point supplied water would likely simply 'fall' through the deep holes that were created by keeping the irrigation points at fixed locations.

[Figure]

**Figure 5:** Hydraulic conductivity, added mass and density over time for 7 individual experiments. In blue the calculated hydraulic conductivity, in red the firn sample mass as a percentage of its mass pre-experiment, and in green the density over time. The dotted line indicates start time of continuous outflow, the dashed line shows the time at which water supply was stopped (note that for experiment *firn2* the 'pumps off'-label is missing, since this experiment was continued for longer than the time displayed here). Grey shading shows where outflow>inflow. The title shows the name of the experiment and its starting date & time [dd-mm, HH:MM], all measurements were carried out in 2021.

**Figure 6**

- *the symbols for the lower and middle air temperature curves are not distinguishable;*
- *the order of the three different air temperature legend entries is counterintuitive;*
- *the order of the air and firn temperature legend entries is counterintuitive, i'd suggest to have air on top in the legend box.*

Please find an updated version of the figure & caption below.

[Figure]

**Figure 6:** Air- and firn temperature over time for experiments *firn1* and *firn3*. The blue dashed line indicates start time of continuous outflow, grey shading shows where outflow>inflow. The black dashed line shows the time at which the experiment was stopped (when pumps were turned off). Coloured lines show temperature evolution at 4 locations within the firn block, ~1 cm above its base. Grey dashed and dotted lines represent air temperatures next to ROSA.

**Figure 13**

- *figures in publications usually have no titles. Their function is taken by the caption.*
- *i'd suggest to "repack" this figure and adopt a structure that is closer to a table: move the references to the right or left so that they make one more column along. The other two columns will be the method and the actual values.*
- *combine the info from Kattelman (1987) by bringing closer together the individual data "pieces": I do not see why Ambach et al 1978 and Vallon et al 1976 need to be wedged in between. This will also allow to get rid of another "dimension" of the figure - grey shading behind references from Kattelman (1987) - in the updated more "table-like" structure that info can be given in the "references" column.*
- *regarding the rows of the "table". The existing structure is logical: methods make the higher order subdivisions, then references can define the lower-level subdivisions with a few grouped by the Kattelman (1987) figure bracket or similar. The original values first reported by this study could be either fitted in this structure or presented as a stand-alone group to make it more obvious what the study's contribution. In either case i think 2 more lines can be presented in this figure: vertical infiltration rates derived at time delay before onset of runoff after the start of spraying divided by the sample thickness (0.17 - 0.44 m / h) and lateral flow speed from dye tracing experiments.*

Thanks for your suggestions. We would propose to keep the figure more or less in it current form and not change it into a table, but have modified it to make it clearer.

[Figure]

**Figure 13:** Flow velocities through snow and **firn** as measured in this study, compared to other values published in literature. Author names preceded by an * indicate that the original papers were not available, quoted values were found in Kattelmann (1987).

**References**

[revised manuscript text omitted]

---

## Author Comment (AC2)

**Response to reviewer 2**

**In situ measurements of meltwater flow through snow and firn in the accumulation zone of the SW Greenland Ice Sheet**

Nicole Clerx, Horst Machguth, Andrew Tedstone, Nicolas Jullien, Nander Wever, Rolf Weingartner, Ole Roessler

Dear Reviewer,

We would like to thank you for your thorough and constructive review of our paper, and the suggestions of how to improve its quality. Below, we respond point by point to all comments, and state how we plan to incorporate them in a revised version of the paper. The responses (normal font style) to the reviewer's comments are written directly into the reviews (displayed in italic font style).

Nicole Clerx,

*Fribourg, June 21, 2022*

**1 General comments**

*1. My general feeling about this paper is that I was very excited about the science that was done, but the paper did not provide adequate discussion (1) about why the results were what they were; (2) about the broader implications of the research; and (3) about assumptions and uncertainties.*

Thanks a lot for your enthusiasm about our study (we are excited about the science too!). We agree that the discussion could be improved, so will make sure it is further refined and expanded in the revised version of the manuscript.

*An example is that the theoretical, modeled lateral flow rates were 3 orders of magnitude less than the observed lateral flow rates. This is a very large discrepancy, but the discussion section does not include discussion of why this large discrepancy might exist. I encourage the authors to work on the discussion section to add more discussion of how their results corroborate or challenge our current understanding of firn meltwater hydrology. It might be useful to provide a simple, qualitative description of how these flow processes operate on different spatial scales based on the results. To add to the implications, can your results add any perspective to our understanding of the fate of firn meltwater; e.g. on longer time scales, what percent of meltwater is running off? Can your results be extrapolated to non-ice slab regions, or are they spatially limited?*

We agree that the discrepancy in calculated and observed flow rates for lateral flow result in significantly different estimates. Similarly, permeability values resulting from having used various methods yield very different results. We will investigate further, and provide a better discussion of how our results fit into the current understanding of meltwater flow and firn hydrology.

Thanks for the suggestion of adding more on the implications of our observations regarding meltwater runoff, we will add more discussion on what our measurements could mean for the overall contribution of runoff to the (surface) mass balance of the Greenland ice sheet.

*The paper does not include uncertainty or error analysis, which I would like to see in a field study like this. I realize it can be difficult to quantify uncertainty in work like this, but even a short section qualitatively describing the uncertainties would improve the paper. For example, does permeability change during percolation with ROSA? How will that affect your quantitative results?*

We agree that this is indeed missing in the current version of the paper. We will include a (better) description and quantification of the uncertainty ranges related to the various measurements in the revised manuscript.

————

*2. A central assumption around your analyses is that the flow is Darcian. It may be the case that this is a valid assumption, but I think it would be appropriate to include justification. Is preferential flow in fingers/pipes Darcian? Would you calculate a different hydraulic conductivity if you conder Richard's Equation vs. Darcy's law? Is there a point at which you expect your Darcian assumption to break down? Your abstract says that for the ROSA experiments, "flow predominantly occurring through preferential flow fingers". Is there a difference in the conductivity between the preferential flow fingers and the matrix-flow instances? Do your calculated hydraulic conductivities (Equation 7 and line 198) represent a bulk' conductivity (that might represent conductivity for matrix flow) or the conductivity in the preferential flow pipes?*

Darcy's law is generally assumed valid as long as flow is linear and laminar (non-turbulent), i.e. having a Reynold's number $Re$ of $<1$, where the Reynold's number is defined as:

$$Re = \frac{\rho u L}{\mu}$$

with $\rho$ is the fluid density [kg m$^{-3}$], $u$ is the flow speed, $L$ is a characteristic linear dimension [m] and $\mu$ is the dynamic viscosity of the fluid [Pa·s]. Darcian flow rates are almost never exceeded in granular materials (Freeze and Cherry, 1979).

We will better specify the underlying assumptions related to the applicaton of Darcy's law and why/whether we think these are valid, and also discuss the differences in hydraulic conductivity values between matrix- and preferential flow in more detail.

————

*3. A structural comment: Consider adding a bit of text at the end of the intro describing the structure of the paper, i.e. outlining, to clarify that there are two distinct but related experiments. You mentioned in the introduction that there were two field seasons/two experiments, but as I read section 4 I kept wondering about the other experiment.*

Thank you for this suggestion, we will incorporate this in the revised manuscript.

————

*4. The slush vs runoff limit finding is mentioned in the abstract, which indicates that it is an important result/outcome of the work. However, the discussion of this is only briefly mentioned at the end seems disparate from the results. I think it would be useful to add a bit more about how this 4 km is calculated and more discussion about the implications, including what additional "essential data" are required.*

We agree. We will include references to sources that we used to determine the 4 km distance that meltwater can flow laterally, and discuss under what assumptions this calculation is valid. We will also include a more expansive discussion of the slush- and runoff limit, to link our findings to the more general hydrologic system on the SW Greenland ice sheet as shown in Fig. 1.

**2 Specific comments**

*Line 89: It may be worth specifying what a 'ripe' snowpack is for TC readers not familiar with snow hydrology.*

Good idea, we will include some explanation in the revised manuscript. (A 'ripe' snowpack means that it has warmed up to 0°C and now consists of metamorphosed, granular snow crystals that can yield meltwater.)

––––––––––

*136: "systematic me asurements of which are required to determine the hydraulic conductivity and water retention capacity of icy firn": this is very vague. Can you specify how they are systematic, and what the parameters are?*

The word "systematic" was used since the original version of ROSA only had an analog flow meter, and inflow was manually steered by opening or closing a plastic valve on a jerry can on top of the device. Hence, flow ratae was was hard to control. In the upgraded ROSA the inflow is governed by a digital flow controller, and water is pumped up actively by aquarium pumps to ensure continuous and constant inflow. We will rephrase this sentence.

––––––––––

*165ish: I would like a bit more detail about the samples here. What are the dimensions of one sample? Are the firn samples taken from different depths in the pit, or side-by-side extractions, (which would allow you to understand spatial variability, perhaps)? Table 1 could include a column that states the depth interval that the firn came from in the pit.*

All samples were roughly 70x70x15 cm in size. The block for the snow experiment originated from a snowpit at ∼1.5 m depth and was made up of older, transformed, relatively coarse-grained snow including layers of depth hoar, alternated with layers of finer-grained wind-blown snow. The firn blocks originated from a 2 m deep quarry close to the laboratory tent at FS4. The samples were extracted side by side, and the depth of their top surface (i.e. the top of the firn layer at the time) was at 1.32 m below the snow surface. We will include this information in the revised manuscript.

––––––––––

*170: Can you be more specific about what is on this checklist?*

The checklist ensures that all relevant metadata of the experiment and the snow/firn sample are recorded, and is quite extensive. We didn't provide more details on what is on the checklist in the manuscript for readability purposes. Categories on the checklist include i.a. date/times (quarrying/transport of the firn block, start/end of the experiment), firn block properties (dimensions, rough stratigraphy, location of and tools used during quarrying, initial weight), and experiment variables (placement location of the various sensors in the sample during the experiment, flow rate).

––––––––––

*201: Is the densification just due to adding mass to the sample, or is there compaction (volume change) too? Also, what is the 'apparent rate of densification'? Is that different than the actual rate?*

The densification is only due to the mass increase – no volume change was observed in the course of the individual experiments. We used the term 'apparent rate of densification' to indicate that this is a transient rate that is valid during the experiments. The 'final' density is not simply the densification rate * experiment duration but less, due to outflow after the water supply was stopped.

*203-205: I found the description of steps here a bit hard to follow. I suggest putting the description into past tense and write as a narrative; e.g. the water flow started, went this long, we observed X, then this happened, etc.*

OK. We will rephrase this sentence.
* * *
*220: Is there a difference between piping and preferential flow? If not just say preferential flow was visible. Also, this is the only instance in the paper in which you use 'piping'; otherwise you use 'fingers', which I think is an interchangeable term. I suggest sticking with a single term.*

Thanks for this remark, there is no intended difference between piping and preferential flow through 'fingers'. In the improved manuscript we will stick to one term.
* * *
*Figure 5 – I suggest coloring the hydraulic conductivity axis (ticks and label) to be blue (same as the dot color) to be consistent with "density" and "added mass" axes. For clarity, I would remove the date-time portion of the subfigure titles, which will make the figure titles consistent with the naming in Table 1 and 2.*

Thanks for these suggestions, we will update the figure accordingly.
* * *
*242: What parameters? I think you say in the next paragraph, but as a reader my initial reaction is that this is vague. I suggest reworking the text a bit to avoid this.*

OK, this is indeed mentioned in the subsequent paragraph. Text will be rephrased.
* * *
*244: This is vague: what is 'shallow'? How deep were the snow pits?*

Both the cores and the snow pits reached the top of the ice slab, which was encountered at a maximum of 1.2 m depth. We will clarify this.
* * *
*246-249: The method here is vague – can you briefly explain the steel-tape method? Are you implying that you remove snow from the hole after drilling the borehole? Doesn't drilling a borehole inherently remove snow?*

The steel-tape method involves chalking the bottom part of a ruler or steel tape that is subsequently lowered into a hole, typically a well, until a certain known depth where the bottom of the tape is below the water table. Upon bringing the tape back to the surface, the wetted part of the chalk indicates the water level. We will include more details on this method in the updated manuscript.
* * *
*275: I think it would help clarify the text if your methods above use the same language of 'slush matrix properties' – it took me a moment to realize that your 'slush matrix properties' described in 5.2.1 were just the properties you were describing in 2nd and 3rd paragraphs of section 5.1. Also – it might be useful for you to include a more formal definition of what you mean by slush matrix.*

Thanks for this suggestion, we'll include a more formal definition of slush and improve the wording in section 5.2.1.

*Section 5.2.2/6.1.2: The discussion of the large variation (a factor of 10) in observed flow velocities is not adequately discussed. Why is there this large variation? Is it just local storage? Snow pack properties?*

Agreed. We think that the main cause for this variation in lateral flow velocity is related to small-scale topography of the ice slab surface over which meltwater flows. We will include this in more detail in the revised version of the manuscript.

*313: Consider adding language like "modeled flow velocities" throughout the text to clearly differentiate between when you are calculating theoretical velocities from an equation and your measurements.*

OK, we will use more consistent language to differentiate calculated and measured velocities.

*Figure 12: I am not sure that this figure is needed, or if you want to include it consider adding a meltwater flux calculation from a surface energy balance model.*

We think this figure is relevant in the description of the field sites providing as background information on the meteorological conditions responsible for the observed meltwater, but it is true that its placement in this part of the paper is suboptimal. We will move it to the section describing the measurement set-up and -location in the revised manuscript.

*335: Can you be a bit more specific about which measurements are comparable? I.e., are the previous measurements that your data agree with capturing the same process?*

Given the variety in age of, and the detail of methodological description in the various papers cited, it is tricky to confidently say to what degree the various measurements are comparable to our data. As far as we are aware, lateral flow velocities through a slush matrix have never been measured. Vertical percolation velocities/firn hydraulic conductivity have been determined before in other studies (i.e. Miller et al. (2018)) although again, either the setting or set-up of the measurements is not always completely analogous. We do agree that comparison of the various values in literature could be better, so we will improve the discussion this section in the revised manuscript.

*340: typo, unsaturated*

OK.

*372: vague sentence – what is relatively large? What is 'more generically'?*

Agreed, will be clarified. We meant to say that we assume that the firn hydrological properties we measured are representative of 'average' firn characteristics in this region of the Greenland ice sheet, and of firn in similar settings (i.e. relatively flat, in the accumulation zone, limited precipitation).

*395: this seems to be a restatement of earlier, but still no why*

Agreed. We will further investigate this matter and provide a better discussion of the differences in permeability values and calculation methods in the updated version of the manuscript.

————————

*447: rewrite sentence – incomplete at this point.*

OK, we will rewrite this sentence.

**References**

Freeze, R. and Cherry, J.: Groundwater, Prentice-Hall, Englewood Cliffs, New Jersey, 1979.

Miller, O., Solomon, D. K., Miège, C., Koenig, L., Forster, R., Schmerr, N., Ligtenberg, S. R. M., and Montgomery, L.: Direct Evidence of Meltwater Flow Within a Firn Aquifer in Southeast Greenland, Geophysical Research Letters, 45, 207–215, https://doi.org/10.1002/2017gl075707, 2018.

---

## Referee Report (RR2)

Review of the updated manuscript

In situ measurements of meltwater flow through snow and firn in the
accumulation zone of the SW Greenland Ice Sheet

by Nicole Clerx, Horst Machguth, Andrew Tedstone, Nicolas Jullien,
Nander Wever, Rolf Weingartner, and Ole Roessler

submitted for publication in The Cryosphere of EGU

Prepared by Sergey Marchenko, postdoctoral research fellow
University of Northern British Columbia and
Vancouver Island University, Canada

In the revised version of the manuscript authors addressed most of the
points I found necessary mentioning in the review of the initial
submission. Please below find a few further suggestions and some
technical corrections.

**Specific comments**

| 1 | L108-109 | Not sure, where the melt rate of 2 mm we per hour comes from? |
|---|----------|---|
| | | Following the suggested assumption of 12 h of melt in one day, that gives 24 mm we of melt per day. With 35 day long melt season that is 840 mm we of melt in a year, which can't be connected with anything cited above. |
| | | One could read ca 600 mm we off Figure 2 in https://tc.copernicus.org/articles/13/1819/2019/ (cited just above) for the annual melt at KAN-U in 2012. That will give 17 mm we per day in summer 2012 on average and 1.4 mm per hour assuming a 12 h melt day... |
| | | And these things can be highly non-linear. But it is also not the aim of this study to assess melt rates. As a potential reader, i'd be pleased with 17 mm per day estimate of max melt rate to be expected as a background info for interpreting the results coming in following chapters. |
| 2 | Chapter 3 when it comes to the Darcy flow law. | Darcy flow lay assumes saturated (one phase) flow. While that is most likely the case for lateral flow described in this study, the vertical flow with extensive fingering is definitely not saturated when it comes to the bulk of the snow/firn mass. I think it is important to mention that here even if in the |

| | | results section the associated calculations are presented as "saturated flow through flow fingers" and the discussion covers the topic in much detail. |
|---|---|---|
| 3 | 247-251, 259-261, headers in Table 2 | To some extent this comment also continues the previous comment.
I would like to encourage authors to be as consistent and pedagogical as possible in using the terms "velocity" and "hydraulic conductivity". Attentive reader surely finds his/her way through the material, but will also appreciate the effort, I believe.

LL389-396 is a good example: formally, there is no mistake, but intuitively it appears as two different properties are compared.

It is not immediately obvious as that flow rate numerically equals hydr. cond. for vertical flow. May be stick with one term (i'd go for velocity, easier to understand) and make a corresponding note.

In L250 area A is, perhaps, meant to be the area of the flow fingers, which is likely not known. That is ok, just make a note on that and possibly speculate that the flow velocity can be a lot higher, if one accounts for that and may be provide a "guesstimate" of what the relative area of fingering flow could have been. |
| 4 | dye tracing of lateral flow | One question arising when interpreting the results from dye tracing experiments is: how long did it take for the tracer to go through the lower gate? The speeds derived using the time first rhodamine portions arrived to the lower gate yield the highest estimate possible. One could have possibly used the times when max R. concentration is observed or the midpoint between first and last portions of the tracer... |

Minor comments

| 1 | L5 | "on the southwest Greenland Ice Sheet". Either southwestERN or add "of the" |
|---|---|---|
| 2 | L25, ref. to Marchenko et al., 2017 | That reference does not quite fit the context it is used in. The article has little to do with firn in Greenland, although it does deal with water percolation. A good reference here could be the RETMIP paper by Vandecrux et al.: https://tc.copernicus.org/articles/14/3785/2020/ - an up to date overview of the reg. scale models at the GrIS and their performance. |

| | | |
|---|---|---|
| | | Where this reference is relevant is the text at LL 127 - 131, since the study showed that the bulk irreducible water content above the flow finger front can be as low as 1% and less as extensive dry (and cold) pockets exist in-between the preferential flow features. |
| 3 | LL44-45 | When reading the definition of the runoff limit, the verb "begins" seems misplaced. May be define runoff limit as the "highest elevation from which runoff occurs", alternatively as "...part of the meltwater present leaves the ice sheet."? |
| 4 | Caption to Fig. 1 | check phrasing of the second sentence. Something is odd there, "eventually" is in the wrong place. |
| 5 | L105 | "and is situated" can be easily skipped |
| 6 | L132 | Consider rephrasing "is analogous to". Water flow through snow literally is flow through porous medium, and ample details on that are given just above) |
| 7 | L189 | is "WT" = water tracer? As a matter of fact, it is nowhere explicitly said that it is. Perhaps good to spell the abbreviation out the first time it is used. |
| 8 | L197-198 | "the distance between the base of the sample and the injector head was constant at 1 m" appearing in parenthesis can be skipped. |
| 9 | L203 | "are": here and throughout the chapter, make sure that tenses are used consistently. So far narration was in the past tense, here we see present, which lower down becomes past again. |
| 10 | L212 | "sensors were inserted into the firn sample. " can be skipped to have "Before the start of each percolation experiment four temperature sensors were inserted horizontally to about 20 cm into the sample ~1 cm above its base." |
| 11 | L262-263 | "…assuming that 2·res equals the average grain size observed in the sample": if that assumption is made, then it is not clear why is the SSA term needed here at all? It does not appear anywhere else, so one may as well get rid of the parameter at all. |
| 12 | LL260-261 and 268-270 | not sure what motivates repetition of the unsaturated flow velocity values. |
| 13 | L309 | "full cloud cover" = overcast? |
| 15 | Equation (12) | is $C_0$ defined later in L325 as "the background conductivity of the meltwater in a borehole…"? Could be good to explicitly define the term. |
| 16 | L345 | "Some residual water…": The irreducible water content can be quantified using the density based parameterization from Schneider and Jansson, 2004 (Journal of Glaciology, Vol. 50, No.168). |

| 17 | L382 | "Hydraulic head variations between individual boreholes throughout the field work period were calculated based on **measured water table heights** along the transect." Are the water table heights determined as described in LL303-304? If yes, then it is most likely of little relevance here, as the water table heights are referenced to the ice slab surface, which can be undulating and highly sloping as stated in ch. 6.3. It is the absolute heights that are important here as they are directly related to the Earth's field of gravity driving the water flow. |
|----|------|---|
| 18 | L384 | "…are relatively high compared to **existing values**": may be "earlier/previously published values" |
| 19 | L506 | "…is the period during meltwater can travel…" add "which" between "during" and "meltwater". |

---

## Author Response (AR2)

**Response to reviewer 1 (Sergey Marchenko)**

**In situ measurements of meltwater flow through snow and firn in the accumulation zone of the SW Greenland Ice Sheet**

Nicole Clerx, Horst Machguth, Andrew Tedstone, Nicolas Jullien, Nander Wever, Rolf Weingartner, Ole Roessler

Dear Reviewer,

We would like to thank you again for your thorough and constructive review of our paper, and the suggestions of how to improve its quality. Below, we respond point by point to all comments, and state how we would incorporate them in a final version of the paper. The responses (normal font style) to the reviewer's comments are written directly into the reviews (displayed in italic font style).

Nicole Clerx,

*Fribourg, September 7, 2022*

**1 Specific comments**

*(1) Not sure, where the melt rate of 2 mm we per hour comes from?*

*Following the suggested assumption of 12 h of melt in one day, that gives 24 mm we of melt per day. With 35 day long melt season that is 840 mm we of melt in a year, which can't be connected with anything cited above.*

*One could read ca 600 mm we off Figure 2 in https://tc.copernicus.org/articles/13/1819/2019/ (cited just above) for the annual melt at KAN-U in 2012. That will give 17 mm we per day in summer 2012 on average and 1.4 mm per hour assuming a 12 h melt day...*

It is exactly from the Figure 2 in the paper by Verjans et al. that you suggest that we got to this value. For 2012, we read off a total surface melt flux of 0.85 m w.e. yr$^{-1}$ from this graph, which with 35 melt days and 12 hours melt per day gives a melt rate of slightly over 2 mm w.e. hr$^{-1}$.

––––––––––

*(2) Darcy flow lay assumes saturated (one phase) flow. While that is most likely the case for lateral flow described in this study, the vertical flow with extensive fingering is definitely not saturated when it comes to the bulk of the snow/firn mass. I think it is important to mention that here even if in the results section the associated calculations are presented as "saturated flow through flow fingers" and the discussion covers the topic in much detail.*

Thank you for this suggestion, we will include a remark in this part of the manuscript as well.

––––––––––

*(3) To some extent this comment also continues the previous comment.*
*I would like to encourage authors to be as consistent and pedagogical as possible in using the terms velocity" and "hydraulic conductivity". Attentive reader surely finds his/her way through the material, but will also appreciate the effort, I believe.*

*LL389-396 is a good example: formally, there is no mistake, but intuitively it appears as two different properties are compared.*

*It is not immediately obvious as that flow rate numerically equals hydr. cond. for vertical flow. May be stick with one term (i'd go for velocity, easier to understand) and make a corresponding note.*

*In L250 area A is, perhaps, meant to be the area of the flow fingers, which is likely not known. That is ok, just make a note on that and possibly speculate that the flow velocity can be a lot higher, if one accounts for that and may be provide a "guesstimate" of what the relative area of fingering flow could have been.*

Thank you for urging us to be as consistent and pedagogical as possible. We believe that in terms of consistency, we have been clear whenever talking about unsaturated percolation velocity or hydraulic conductivity (saturated flow velocity). In the methods-section we believe our explanation of the can not be significantly improved without compromising clarity and conciseness.

Regarding your suggestion on LL389-396: we think it is important to leave the phrasing as it is currently, to emphasize the fact that we are effectively comparing measurements that were done using different techniques, for different goals: in some cases it was merely the percolation velocity measurements that were the goal, other publications focused on

quantifying the material properties of the studied snow and firn and hence present real'
hydraulic conductivity values.

As you say, in L250 $A$ is indeed the total surface of the firn sample whereas it would
be more appropriate to take the area of the flow fingers (which is impossible because
we weren't able to measure the surface area of the preferential flow paths). We will
incorporate a comment on this in the final version of the manuscript, including a remark
regarding the potential underestimation of the flow velocity.

———————

*(4) One question arising when interpreting the results from dye tracing experiments is:
how long did it take for the tracer to go through the lower gate? The speeds derived using
the time first rhodamine portions arrived to the lower gate yield the highest estimate
possible. One could have possibly used the times when max R. concentration is observed
or the midpoint between first and last portions of the tracer...*

Given the simplistic nature of our RWT experiments (visual observation of rhodamine
displacement), quantification of RWT concentrations was not possible. For future exper-
iments it would indeed be better to use the maximum concentration instead of the first
arrivals, but this wasn't feasible at the time.

**2 Minor comments**

*(1) L5: "on the southwest Greenland Ice Sheet". Either southwestERN or add "of the"*

OK.

———————

*(2) L25, ref. to Marchenko et al., 2017: That reference does not quite fit the context it
is used in. The article has little to do with firn in Greenland, although it does deal with
water percolation. A good reference here could be the RETMIP paper by Vandecrux et
al.: https://tc.copernicus.org/articles/14/3785/2020/ - an up to date overview of the reg.
scale models at the GrIS and their performance.*

*Where this reference is relevant is the text at LL127 - 131, since the study showed that
the bulk irreducible water content above the flow finger front can be as low as 1% and less
as extensive dry (and cold) pockets exist in-between the preferential flow features.*

Thanks for these comments. We will change the original citation to the RETMIP-paper,
and include an additional comment + reference to the paper quoted initially.

———————

*(3) LL44-45: When reading the definition of the runoff limit, the verb "begins" seems
misplaced. May be define runoff limit as the "highest elevation from which runoff occurs",
alternatively as "...part of the meltwater present leaves the ice sheet."?*

This was the initial definition that we had thought of, but after the various observed melt
seasons in the area we realised that it is not certain that (all) meltwater present leaves the
ice sheet: refreezing can occur during/after colder episodes, even below the runoff limit.
Hence we would propose to stick to the current formulation.

———————

*(4) Caption to Fig. 1: check phrasing of the second sentence. Something is odd there,
"eventually" is in the wrong place.*

OK.

*(5) L105: "and is situated" can be easily skipped*

OK.

———————

*(6) L132: Consider rephrasing "is analogous to". Water flow through snow literally is flow through porous medium, and ample details on that are given just above)*

Indeed. We will rephrase this sentence.

———————

*(7) L189: is "WT" = water tracer? As a matter of fact, it is nowhere explicitly said that it is. Perhaps good to spell the abbreviation out the first time it is used.*

Correct, WT is the abbreviation for water tracer. We will include the full definition of the abbreviation in the manuscript.

———————

*(8) L197-198: "the distance between the base of the sample and the injector head was constant at 1 m" appearing in parenthesis can be skipped.*

OK.

———————

*(9) L203: "are": here and throughout the chapter, make sure that tenses are used consistently. So far narration was in the past tense, here we see present, which lower down becomes past again.*

We will check and homogenize the tense(s) used.

———————

*(10) L212: "sensors were inserted into the firn sample." can be skipped to have "Before the start of each percolation experiment four temperature sensors were inserted horizontally to about 20 cm into the sample ~1 cm above its base.*

OK.

———————

*(11) L262-263: "...assuming that 2·res equals the average grain size observed in the sample": if that assumption is made, then it is not clear why is the SSA term needed here at all? It does not appear anywhere else, so one may as well get rid of the parameter at all.*

We included the SSA-term for completeness and in particular to adhere to the original definition of the equation by Calonne et al.

———————

*(12) LL260-261 and 268-270: not sure what motivates repetition of the unsaturated flow velocity values.*

Thanks, we adjusted this to remove the repetition.

———————

*(13) L309: "full cloud cover" = overcast?*

Correct, we intentionally use the term "full cloud cover" here since it is more specific than overcast (which can mean partially cloud covered as well).

*(15) Equation (12): is $C_0$ defined later in L325 as "the background conductivity of the meltwater in a borehole..."? Could be good to explicitly define the term.*

The reason that "background conductivity" is only mentioned later is because the equation is valid for any type of tracer and hence the fact that it concerns background conductivity in this case is experiment-specific.
* * *
*(16) "Some residual water...": The irreducible water content can be quantified using the density based parameterization from Schneider and Jansson, 2004 Journal of Glaciology, Vol. 50, No.168).*

Yes, if one would know all the parameters in the equation they use:
$$S_{wi} = \frac{\theta_{mi} \frac{\rho_f}{\rho_w}}{\phi}$$
With $S_{wi}$ residual water saturation [-], $\theta_{mi}$ irreducible liquid water content [-], $\rho_f$ sample density [kg m$^{-3}$], $\rho_w$ water density [kg m$^{-3}$] and $\phi$ matrix porosity [-]. In our case, it was unfortunately impossible to determine the irreducible liquid water content exactly, given that we don't know the difference in density of the matrix before and after full saturation.
* * *
*(17) L382: "Hydraulic head variations between individual boreholes throughout the field work period were calculated based on measured water table heights along the transect." Are the water table heights determined as described in LL303-304? If yes, then it is most likely of little relevance here, as the water table heights are referenced to the ice slab surface, which can be undulating and highly sloping as stated in ch. 6.3. It is the absolute heights that are important here as they are directly related to the Earth's field of gravity driving the water flow.*

Correct. However, we decided to stick to our recorded values for water table height referenced to the ice slab surface. For clarity, but principally because we don't have a DEM at our disposition that is accurate enough at the scale of our field measurements to convert these water table depths (and snow heights) to their absolute 'height'.
* * *
*(18) L384: "...are relatively high compared to existing values": may be "earlier/previously published values*

Thanks, we will rephrase this.
* * *
*(19) L506: "...is the period during meltwater can travel..." add "which" between "during" and "meltwater".*

OK.

**Response to reviewer 2**

**In situ measurements of meltwater flow through snow and firn in the accumulation zone of the SW Greenland Ice Sheet**

Nicole Clerx, Horst Machguth, Andrew Tedstone, Nicolas Jullien, Nander Wever, Rolf Weingartner, Ole Roessler

Dear Reviewer,

We would like to thank you again for your thorough and constructive review of our paper, and the technical comments you made for improving the quality of its final version. We have directly incorporated your comments into the revised manuscript.

Nicole Clerx,

*Fribourg, September 7, 2022*

---

## Author Response (AR3)

**Response to editor comments**

**In situ measurements of meltwater flow through snow and firn in the accumulation zone of the SW Greenland Ice Sheet**

Nicole Clerx, Horst Machguth, Andrew Tedstone, Nicolas Jullien, Nander Wever, Rolf Weingartner, Ole Roessler

Dear Kirstin Poinar,

We would like to thank you for your editing efforts and constructive review of our paper. Please find a response to your comments below.

Nicole Clerx,

*Fribourg, September 20, 2022*
* * *
*A. line 253 "could be a lot higher" – please provide an estimate (a rough estimate is okay) in place of "a lot".*

Based on minimum values of 'area occupied by preferential flow paths as a percentage of total snow/firn sample area' (5–30% as reported by Williams et al., 2010; Katsushima et al., 2013), we have included a estimate of the (maximum) potential percolation velocities in our experiments.
* * *
*B. line 335 – Reviewer 1 commented that the timing is biased low, as the authors used time of first arrival, rather than time of peak concentration through. In their response, the authors pointed out that they did not measure concentrations (it was not possible with the methods/equipment they had), so the peak time could not be discerned. Nevertheless, the authors should comment on this timing bias —- it means their velocities are biased high. Comment here or in the discussion how this compounds or offsets other sources of error.*

OK, we will include a comment here that our simple measurements represent high estimates of the lateral flow velocity.
* * *
*C. Reviewer 1 points out that the datum for the water table is the ice slab surface, which can undulate locally and is not likely to be level or even flat. The authors should comment on this – probably in the discussion (lines 466-7), or alternately lines 307-8, where the water table measurement method is described, or lines 386-8. Because the authors are concluding no correlation between the local water table height and lateral flow velocity (stated on line 466), these undulations may well matter, and should be addressed.*

Agreed. We will elaborate on the potentially varying reference height for the water depth measurements due to the ice slab undulations.
* * *
*D. Table 2 - unsaturated has no quotes, while 'saturated' has quotes. Why? Suggest to choose a style and apply to both for consistency.*

OK. We will update the table header.

**References**

Katsushima, T., Yamaguchi, S., Kumakura, T., and Sato, A.: Experimental analysis of preferential flow in dry snowpack, Cold Regions Science and Technology, 85, 206–216, https://doi.org/10.1016/j.coldregions.2012.09.012, 2013.

Williams, M. W., Erickson, T. A., and Petrzelka, J. L.: Visualizing meltwater flow through snow at the centimetre-to-metre scale using a snow guillotine, Hydrological Processes, pp. n/a–n/a, https://doi.org/10.1002/hyp.7630, 2010.